# Efficient overall water splitting in acid with anisotropic metal nanosheets

Dongshuang Wu[1✉], Kohei Kusada [1✉], Satoru Yoshioka[2], Tomokazu Yamamoto[2], Takaaki Toriyama[3], Syo Matsumura[2,3], Yanna Chen[4], Okkyun Seo[4], Jaemyung Kim [4], Chulho Song[4], Satoshi Hiroi[4], Osami Sakata [4], Toshiaki Ina[5], Shogo Kawaguchi [5], Yoshiki Kubota [6], Hirokazu Kobayashi[1] & Hiroshi Kitagawa [1✉]

Water is the only available fossil-free source of hydrogen. Splitting water electrochemically is among the most used techniques, however, it accounts for only 4% of global hydrogen production. One of the reasons is the high cost and low performance of catalysts promoting the oxygen evolution reaction (OER). Here, we report a highly efficient catalyst in acid, that is, solid-solution Ru–Ir nanosized-coral (RuIr-NC) consisting of 3 nm-thick sheets with only 6 at.% Ir. Among OER catalysts, RuIr-NC shows the highest intrinsic activity and stability. A home-made overall water splitting cell using RuIr-NC as both electrodes can reach 10 mA cm$^{-2}_{geo}$ at 1.485 V for 120 h without noticeable degradation, which outperforms known cells. Operando spectroscopy and atomic-resolution electron microscopy indicate that the high-performance results from the ability of the preferentially exposed {0001} facets to resist the formation of dissolvable metal oxides and to transform ephemeral Ru into a long-lived catalyst.

[1] Division of Chemistry, Graduate School of Science, Kyoto University, Kyoto, Japan. [2] Department of Applied Quantum Physics and Nuclear Engineering, Kyushu University, Fukuoka, Japan. [3] The Ultramicroscopy Research Center, Kyushu University, Fukuoka, Japan. [4] Synchrotron X-ray Group and Synchrotron X-ray Station at SPring-8, National Institute for Materials Science, Hyogo, Japan. [5] Research and Utilization Division, Japan Synchrotron Radiation Research Institute (JASRI), Hyogo, Japan. [6] Department of Physical Science, Graduate School of Science, Osaka Prefecture University, Osaka, Japan. ✉email: dongshuangwu@kuchem.kyoto-u.ac.jp; kusada@kuchem.kyoto-u.ac.jp; kitagawa@kuchem.kyoto-u.ac.jp

Water electrolysis is one of the most feasible ways to provide hydrogen in the establishment of clean and renewable energy cycles, accounting around 4% global hydrogen production[1,2]. Water electrolysis involves two half-cell reactions: cathodic hydrogen evolution reaction (HER) to generate hydrogen and anodic OER to generate oxygen. HER is kinetically favoured in acid because of the high proton concentration, where current densities such as $10 \, mA \, cm^{-2}_{geo}$ are readily achieved even at (tens of) millivolts of overpotential with a negligible amount of Pt catalysts[3]. The recent successes in solid polymer exchange membrane technology further accelerate water electrolysis in acid instead of conventional alkaline media[4–6]. However, the critical bottleneck lies in OER, particularly in acidic media, because most metals dissolve in the working potential region according to Pourbaix diagrams[2,7]. OER is an uphill energy transformation process involving four-proton and -electron transfers per oxygen molecule, which typically requires catalysts to work under high overpotentials and counteracts the long-term operation of catalysts. Therefore, delivering a high current density at a reduced overpotential is essential to the durability of catalysts. Although several principles have been proposed for designing OER catalysts for use in alkaline media[8–11], there are only a few reports referred to the design principles for the catalysts used in acid[12]. Currently, only Ir oxides show moderate stability for OER in acid[5,6,13,14], but still require high overpotentials (generally over 300 mV). In contrast, Ru is the most active OER catalyst and is nearly 5–16 times cheaper than Ir these 5 years, however, Ru has a serious degradation problem[15–19]. The prevailed methods to improve the stability of Ru catalysts, such as heavy Ir doping (at least require 30 at.% of Ir)[20], using thermal calcination[21,22] and strong support-metal interaction effect[23], are generally at the sacrifice of activity. Exploring non-degradable OER catalysts capable of operating in acid at low overpotential is one of the greatest challenges in current electrochemistry field.

Here, we report a highly active and stable Ru-Ir catalyst for overall water splitting in acid. Atomic-resolution electron microscopy and 3D tomography revealed that the obtained Ru-Ir catalyst had a unique coral-like structure consisting of 3-nm-thick sheets with extended {0001} facets and composed with only 6 at.% Ir (denoted as **RuIr-NC**). **RuIr-NC** showed the lowest overpotential (165 mV) to achieve $10 \, mA \, cm^{-2}_{geo}$ for OER in acid. Moreover, **RuIr-NC** achieves high mass activity and specific activity, which are 1–2 orders of magnitude higher than the reported highly active catalysts. The obtained **RuIr-NC** is also matched with remarkable stability that it showed no noticeable degradation throughout 122 h under a fixed geometric current density of $1 \, mA \, cm^{-2}$. In contrast, the comparative OER testing using the spherical RuIr catalyst shows an ordinary degradable behaviour within 1 h. Moreover, **RuIr-NC** displays comparable activity with commercial Pt/C toward HER. A home-made overall water-splitting cell using **RuIr-NC** as both electrodes achieves $10 \, mA \, cm^{-2}$ water-splitting current density with only 1.485 V and keeps operating over 120 h without obvious degradation, which outperforms the known cells in acidic media. One should note that the as-built overall splitting cell is less expensive than the combination of commercial Pt and $IrO_x$ catalysts. Combining operando spectroscopy and atomic-resolution electron microscopy, we revealed that the coral-like structure with exposed extended {0001} facets is the exclusive factor that contributes the enhanced activity and stability. Although metal-based catalyst, perticularly metal nanomaterials, has long been considered less stable for OER, our result shows the possibility of exploring highly efficient and stable water-splitting catalyst by optimizing the structure of the metal. Given the well-established database for controlling the structure and shape of binary or even multiple metallic nanomaterials during the past decades, our finding points out the possibility of a fast-screening process to access active and stable catalysts by using the database.

## Results and discussion

**Catalyst synthesis and general characterization.** **RuIr-NC** was obtained by adding a mixture of $RuCl_3 \cdot nH_2O$ and $H_2IrCl_6$ aqueous solutions to triethylene glycol (TEG) solution containing polyvinylpyrrolidone (PVP) at 230 °C. Transmission electron microscopy (TEM) image shows that the mean diameter of the obtained particles is 57 ± 7 nm (Fig. 1a). We further probed the atomic structure of **RuIr-NC** by an aberration-corrected scanning TEM (STEM) coupled with energy-dispersive X-ray spectroscopy (EDS). The high-angle annular dark-field (HAADF)-STEM image in Fig. 1b shows that the particle consists of sheet-like fragments. The 3-D tomography (Fig. 1d, Supplementary Fig. 1 and Movie 1) further confirms that the particle is assembled by anisotropically grown 2D nanosheets forming a coral-like shape. The nanosheets expose a hexagonal atomic arrangement corresponding to the (0001) hexagonal closed-packed (hcp) lattice plane (Fig. 1c and Supplementary Fig. 2). Rietveld refinement of the synchrotron X-ray diffraction (XRD) pattern also confirmed the anisotropic growth of **RuIr-NC** with an extended (0001) plane (Supplementary Fig. 3). The crystal sizes estimated from the sharp peaks of (100) and (110) are 15.2(3) and 13.4(5) nm, respectively, while that from (0002) was 3.1(2) nm (Supplementary Fig. 4). The difference in the crystal size confirmed the extended (0001) plane in **RuIr-NC**, which is consistent with the STEM observation.

A selected area in Fig. 1c indicated by the white rectangle was further analysed to provide Ir position in the sheet and the sheet thickness (Fig. 1e). The colour represents the Gaussian peak intensity extracted from the corresponding atomic column position. The three lines denoted as I–III in Fig. 1e were quantitatively analysed to give the numbers of Ir and Ru atoms (Fig. 1f). The total number of atoms in one atomic column ranges between 6 and 9, corresponding to ~3-nm-thick sheets, which agrees well with the thickness measured from the HAADF-STEM image (Supplementary Fig. 9) and estimated crystal size from XRD (Supplementary Fig. 4). The atomic column positions containing 1 or 2 Ir atoms are denoted by the arrows (Fig. 1e), which clearly shows the random distribution of Ir atoms in the Ru hcp lattice. This result suggests the formation of a Ru-Ir solid-solution. EDS maps (Fig. 1g–i) demonstrate the homogeneous distribution of Ir and Ru atoms throughout the entire particle, which is consistent with the analyses in Fig. 1e, f. In addition, the atomic ratio of Ru and Ir estimated from EDS spectra was 0.96:0.04 (one spectrum corresponds to the particle shown in Fig. 1b was shown in Supplementary Fig. 10), which is in line with the X-ray fluorescence spectroscopy result (0.94:0.06).

As references for the catalytic tests, Ru nanoparticles (NPs), Ir NPs and Ru-Ir nanospherical particles (**RuIr-NS**) with the size of 4.9 ± 0.8, 1.9 ± 0.3 nm and 3.9 ± 0.7, respectively, were prepared (Supplementary Figs. 3, 5–8). **RuIr-NS** was synthesized by heat-up method, that is heating the mixture of metal precursors in TEG and PVP solution from RT to 230 °C. In contrast, **RuIr-NC** is obtained through a hot-injection process. It is known that the rate of surface atom diffusion is very high at an elevated temperature, which prompts the exposure of facet with lower surface energy[24]. Therefore, the hot-injection method would realize **RuIr-NC** mainly exposing the lowest energy (0001) facets[25]. **RuIr-NS** was confirmed to have the same atomic ratio and solid-solution nature as that of **RuIr-NC** (Supplementary Fig. 11).

**Comparison of OER activity and stability.** The OER performance of **RuIr-NC** was investigated in a $0.05 \, M \, H_2SO_4$ aqueous

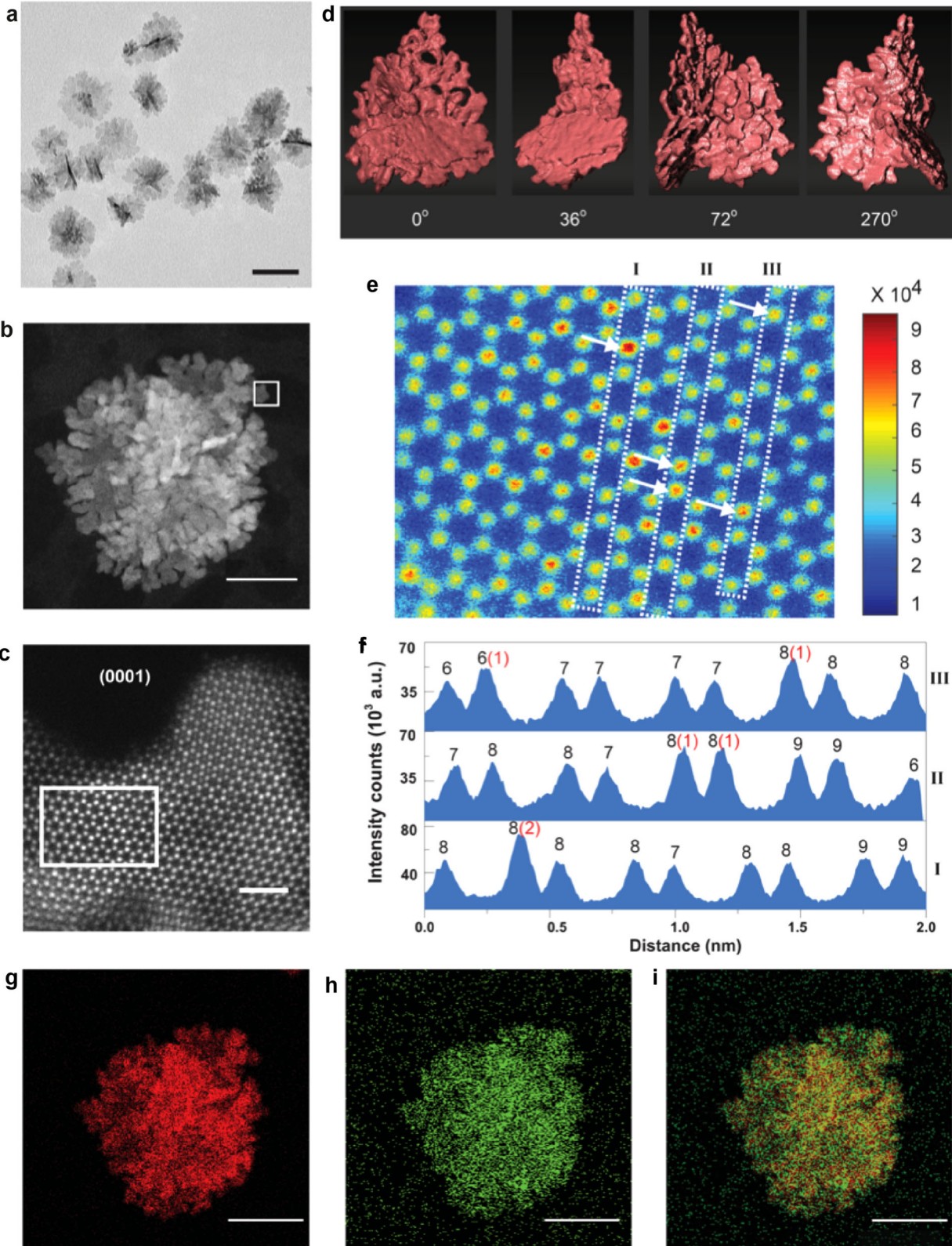

**Fig. 1 Morphological and structural characterization of RuIr-NC. a** Transmission electron microscopy (TEM) and **b** high-angle annular dark-field (HAADF)-scanning TEM image. **c** Atomic-resolution HAADF-STEM image from the selected area in (**b**) showing the extended (0001) plane. **d** 3D tomographic reconstruction of **RuIr-NC**. **e** The magnified image of the selected area in (**c**), where the colour bar represents the Gaussian intensity of the electron diffraction located at the column position. **f** Quantitative analysis of the Ru and Ir distribution at lines I–III in (**e**). The black and red numbers above each peak represent the number of total and Ir atoms in the corresponding position, respectively, and the Ir positions are noted by arrows in (**e**). Elemental maps showing **g** Ru-*L* (red), **h** Ir-*M* (green) and **i** the overlay of **RuIr-NC** in (**b**). The scale bars in (**a**), (**b**), (**c**), and (**g–i**) are 50, 20, 1, and 25 nm, respectively.

solution using a rotating disk electrode (RDE) by linear sweep voltammetry (LSV) in a procedure similar to that used in benchmark studies[3,6]. All the LSV curves are background current and ohmic corrected and the currents contributed from the bare RDE and carbon/Nafion mixture at higher potentials are negligible (Supplementary Fig. 12).

We compare the OER activity in three prevailed metrics, i.e, geometric activity ($j_{geo}$, geometric area of RDE), mass activity ($j_m$, mass of metals) and specific activity ($j_s$, electrochemical active surface area (EASA)). Geometric activity is important on the device level. With the same catalysts loading amount, **RuIr-NC** requires only 165 mV to achieve 10 mA cm$^{-2}_{geo}$ (Fig. 2a), showing much higher activity than Ir NPs (371 mV) and Ru NPs (550 mV), and the known optimal catalysts in acid (Supplementary Table 1). We must note that to avoid the overestimation of OER current due to the metal oxidation, we performed LSV from high potential to low potential (cathodic scan). Scan positively exerts a more serious oxidation/dissolution on the metals. However, the high OER activity/stability of **RuIr-NC** can be also reflected by the almost overlapped anodic and cathodic LSVs (Supplementary Fig. 13). The Faradaic efficiency of the **RuIr-NC** was 98.5% at 20 mA cm$^{-2}$ which suggests that the oxidation current is mainly derived from OER (Supplementary Fig. 14). To put the performance of **RuIr-NC** into perspective, we compared its mass activity with the that of representatively active catalysts listed in Supplementary Table 1. **RuIr-NC** is two magnitudes higher in $j_m$ than those of the benchmark IrO$_2$ and RuO$_2$ NP[26] (Fig. 2b). Specifically, at 1.45 V, **RuIr-NC** shows a $j_m$ of 796 A g$^{-1}_{metal}$ which is 2–4 times higher than the reported highly active catalysts[6,22,26,27]. The specific activity is known to be fundamentally determined by the catalysts' EASA. The specific EASA (normalized by mass) of **RuIr-NC** was checked by using Cu underpotential deposition (UPD) method (Supplementary Fig. 15a–c). Remarkably, we found that at 1.45 V, the **RuIr-NC** showed a $j_s$ of 4.4 mA cm$^{-2}$ (Supplementary Fig. 15d), which is approximately one magnitude higher than the reported highly active catalysts with considerable stability such as IrO$_x$/SrIrO$_3$ (ca. 0.67 A cm$^{-2}$)[5]. From these activity evaluations, in both device-specific and material-specific figure-of-merits, **RuIr-NC** is no doubt the most active catalyst for OER in acid, i.e., achieving a high current density with a low overpotential.

The highly efficient **RuIr-NC** also combined a remarkable long-term catalytic stability. Ru NPs quickly lost the activity and showed negligible current density after the 1st LSV scan (Supplementary Fig. 16a), which is consistent with a previous study[19]. To evaluate the stability of Ir NPs and **RuIr-NC**, we adopted the chronopotentiometry (CP) measurement under a constant current density of 1 mA cm$^{-2}_{geo}$. As shown in Fig. 2c, a sharp rise in electrode potential indicating severe degradation was observed within 12 h for Ir NPs. In contrast, the electrode potential of **RuIr-NC** showed no noticeable change throughout 122 h. The **RuIr-NC** can sustain 40 h even under 10 mA cm$^{-2}_{geo}$ (Supplementary Fig. 17). Such long-term durability is unachieved among known Ru metal-based OER catalysts (Supplementary Table 1). Therefore, the **RuIr-NC** outperforms the best state-of-the-art OER catalyst in acidic media with a remarkably reduced overpotential and long-term stability, representing a competitive catalyst of metal-based nanomaterials.

As **RuIr-NC** is more active and durable for OER than the typical metal-based OER catalysts, we assumed that it was contributed by the unique nanocoral structure. This assumption is verified by comparing its catalytic performance with that of ordinary **RuIr-NS**. At 10 mA cm$^{-2}_{geo}$, **RuIr-NS** requires an overpotential of 242 mV (Fig. 2a), which is nearly 77 mV higher than that of **RuIr-NC** (165 mV). However, when scan positively, **RuIr-NS** showed a quite similar or slightly higher current density

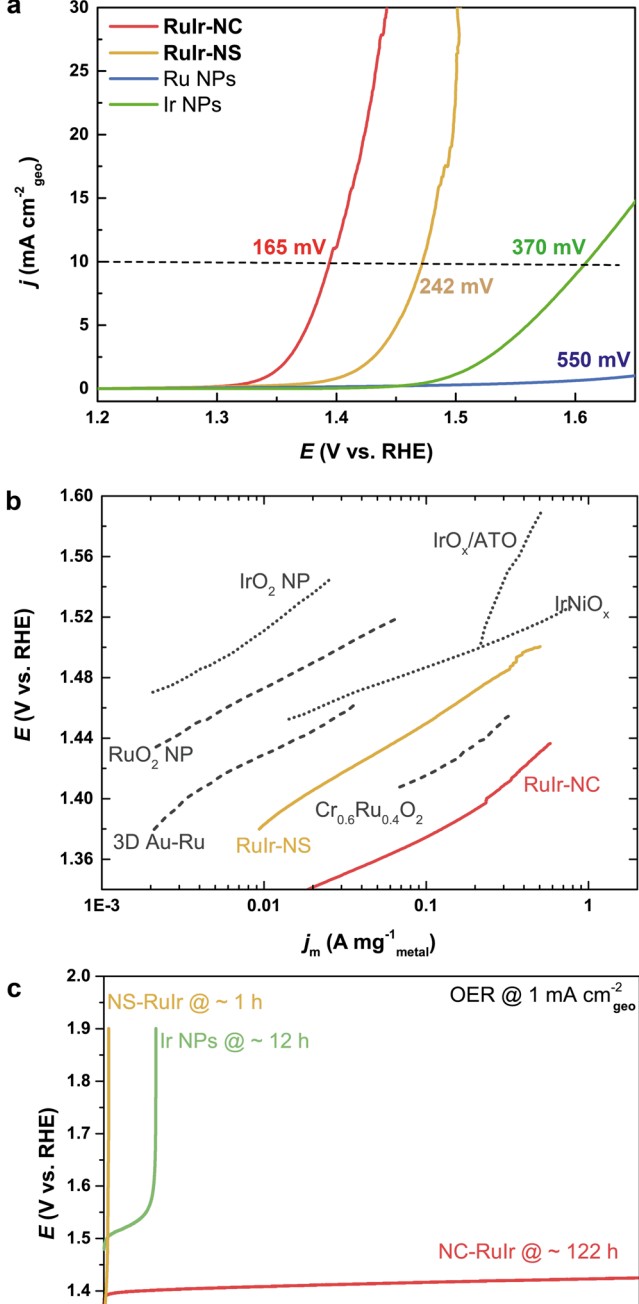

**Fig. 2 Oxygen evolution reaction (OER) performance. a** Geometric activity (current density normalized by electrode surface area) of RuIr catalysts (red for **RuIr-NC** and orange for **RuIr-NS**) and pure Ru NPs (blue) and Ir NPs (green) (LSV profiles are obtained by cathodic scan). **b** Mass activity of **RuIr-NC**, **RuIr-NS** and the reported high-performance catalysts with considerable stability in acid including RuO$_2$ NP[26], IrO$_2$ NP[26], IrO$_x$-ATO[23], 3D Au–Ru[46], IrNiO$_x$[6] and Cr$_{0.6}$Ru$_{0.4}$O$_2$[27]. The mass activities are normalized by the mass of noble metals. **c** Chronopotentiometric curves under the OER current density of 1 mA cm$^{-2}_{geo}$.

than **RuIr-NC** before 1.4 V, and its current density did not increase as sharply as that of **RuIr-NC** thereafter (Supplementary Fig. 13a). This suggests the serious metal oxidation/dissolution in the **RuIr-NS**. The stability test further shows that the activity of

**RuIr-NS** quickly deteriorated within 1 h (Fig. 2c), which was found in typical Ru-based catalysts[17,19].

Inductively coupled plasma mass spectrometry was used to monitor the dissolution of Ru and Ir of the RuIr catalysts to valid our speculation. Almost 95 and 85% of Ru and Ir, respectively, of the **RuIr-NS** dissolved after five repeated polarization scans. In contrast, **RuIr-NC** did not dissolve as much after the second scan and had only 25% and 15% of Ru and Ir loss, respectively, even after five scans (Supplementary Fig. 16b–d). These comparative electrochemistry studies strongly demonstrated that the unique nanocoral structure is highly resistant to the dissolution during OER compared to its spherical counterpart.

**Ex situ and operando characterization of** RuIr-NC **for OER**. To explore the mechanism of the significantly enhanced OER performance of **RuIr-NC** compared to **RuIr-NS**, comprehensive spectroscopies were adopted to investigate the effect of the differences in the electronic structure and crystal size. They are generally considered to contribute to the OER performance. Hard X-ray photoelectron spectroscopy (HAXPES) with a probe depth around 20 nm was performed to give the bulk electronic structure of the RuIr catalysts (Supplementary Fig. 18). The as-prepared **RuIr-NC** and **RuIr-NS** show exactly overlapped Ru3$p$ and Ir4$f$ core-level spectra (Supplementary Fig. 18a, b), suggesting that they have a similar electronic structure. Lab XPS results show that the surface composition and electronic structure of Ru and Ir in the two RuIr catalysts are similar as well (Supplementary Fig. 19 and Table 4). The X-ray absorption fine structure (XAFS) was conducted to obtain the local structure of Ir in RuIr catalyst. Both the X-ray absorption near-edge spectroscopy (XANES) (Supplementary Fig. 20a) and extended XAFS (EXAFS) in Ir $L_3$-edge (Supplementary Fig. 20b–d and Table 2) suggest that, compared with **RuIr-NS**, **RuIr-NC** is not supposed to have special Ir sites. Furthermore, the influence of crystal size effects was also excluded. An isotropic Ru-Ir catalyst was prepared by a similar heat-up method to that of **RuIr-NS** and subsequent high-temperature annealing under vacuum (Marked as **RuIr-L**). It has a similar crystal size, EASA and atomic ratio to **RuIr-NC** (Supplementary Figs. 21 and 22). However, it shows even lower geometric activity and similar degradation behaviour to **RuIr-NS** (Supplementary Fig. 23). **RuIr-NC** and **RuIr-NS** have different crystal size. However, the performance of **RuIr-L** could suggest that the crystal size might not be a determined reason within the study of interest. These measurements collectively suggest that the unique coral structure is the exclusive reason for the enhanced catalytic stability of **RuIr-NC**.

To obtain additional direct evidence of the superior stability of **RuIr-NC**, we conducted operando XANES to both qualitatively and quantitatively monitor the structural change in the RuIr catalysts (setup in Supplementary Fig. 24). Supplementary Fig. 25 shows Ru $K$-edge XANES data from open circuit potential (OCP) jumped to 1.25 V and then to 1.80 V with every 0.05 V intervals. For both catalysts, the absorption edge shifts from Ru towards RuO$_2$ with increasing potential, suggesting an increasing oxide percentage. By using a principal component analysis algorithm and target transformation (Supplementary Figs. 26 and 27), the possible species contained in these XANES spectra were determined as two components, i.e., Ru metal and RuO$_2$. Next, linear combination fitting (LCF) was used to evaluate the proportions of the two components (Supplementary Fig. 28 and Table 3) in order to provide a quantitative understanding of the oxidation rate. Figure 3a, b plot the percentages of Ru and RuO$_2$ component as a function of potential during OER process of **RuIr-NS** and **RuIr-NC**, respectively. The two catalysts show quite similar XANES features below ca.1.5 V (Supplementary

Fig. 29a–c), which is in accordance with their comparable activity below 1.4 V when scanned positively (Supplementary Fig. 13a). This result implies that they have similar OER-active surface structures of -O, or -OOH[15,28,29]. However, this tendency changed thereafter. After 1.5 V, the oxidation of **RuIr-NS** is sharply accelerated and only ca. 25% metallic component remained at 1.8 V, whereas the oxidation of **RuIr-NC** was much slower and 60% metallic component was maintained at 1.8 V (Fig. 3a, b, and Supplementary Fig. 29d). After the operando XANES measurement, although both catalysts show only the original hcp alloy XRD pattern (Supplementary Fig. 30), core-level Ru3$p$ and Ir4$f$ spectra obtained by HAXPES verified the formation of oxides (Supplementary Fig. 18c, d). These results suggest that the metal oxides formed in both catalysts are amorphous. The Ru/Ir compositions of the catalysts during OER were measured by EDX (Supplementary Table 4). Before 1.4 V, the Ru/Ir compositions in **RuIr-NS** and **RuIr-NC** samples were not changed so much. However, the percentage of Ir in **RuIr-NS** much more increased at 1.4 V. In contrast, the percentage of Ru and Ir in **RuIr-NC** did not change obviously even at 1.8 V. This implied that the dissolution rate of Ru in **RuIr-NS** is much higher than that of **RuIr-NC**. We also monitored the change of Ir $L3$-edge XANES during OER and found that the Ir in **RuIr-NC** is more resistant to oxidation compared that in **RuIr-NS**, which is similar to of the result of Ru $K$-edge (see detailed discussion in Supplementary Fig. 20e, f). To elucidate the atomically local structure of the catalysts during OER, we measured HAADF-STEM images at three potentials:1.25 V (just prior to OER), 1.40 V (around the cross-point of the LSVs of the two RuIr catalysts, Supplementary Fig. 13a) and 1.80 V (end potential). The large area TEM images clearly demonstrate the decrease in the number and size of particles of **RuIr-NS** with increasing potential (Supplementary Fig. 31), which suggests the severe dissolution of **RuIr-NS**. The atomic-resolution HAADF-STEM images shown in Fig. 3c–e show that some parts of the remaining particles of **RuIr-NS** still show a crystalline metal lattice; however, most of the particles are entirely amorphous (hydro)oxides. In contrast, for **RuIr-NC**, the coral-like shape showing the hexagonal lattice of the hcp (0001) plane is mainly observed up to 1.80 V and size distribution of the catalyst is maintained (Fig. 3f–h and Supplementary Fig. 32). Notably, the HAADF-STEM image (Supplementary Fig. 33) shows that the growth of the amorphous oxide is affected by the metal facets: there is ca. 1-nm-thick amorphous part along with the [0001] direction whereas there is ca. 5-nm-thick along with the [10$\bar{1}$0] direction. This type of phenomenon is observed by electron diffraction and Auger spectroscopy studies on single-crystalline Ru(0001) and (100) surfaces under electrochemical and gas-phase conditions[30]. In addition, the (0001) facet has the lowest surface energy of hcp structures[25], and thus, it would better resist oxidation than other facets. CV of **RuIr-NC** in Ar-saturated 0.05 M H$_2$SO$_4$ shows similar feature to that of single-crystalline Ru(0001) electrode (Supplementary Fig. 34), which is consistent with the STEM analyses and verifies that (0001) facet of hcp structure might protect Ru from corrosion. From the operando and ex situ measurements, the significant enhanced OER performance of **RuIr-NC**, can be attributed from the exposed extended {0001} facets of the unique coral structure.

**HER and overall water-splitting performance**. Bifunctional catalysts for both HER and OER are highly desirable for practical use as an integrated water electrolyzer. In addition, Ru and Ir metals are also reported to catalyse HER[31]. The HER properties of **RuIr-NC** were investigated by using RDE in the same electrolyte as OER. The overpotential at 10 mA cm$^{-2}_{\text{geo}}$ of **RuIr-NC**

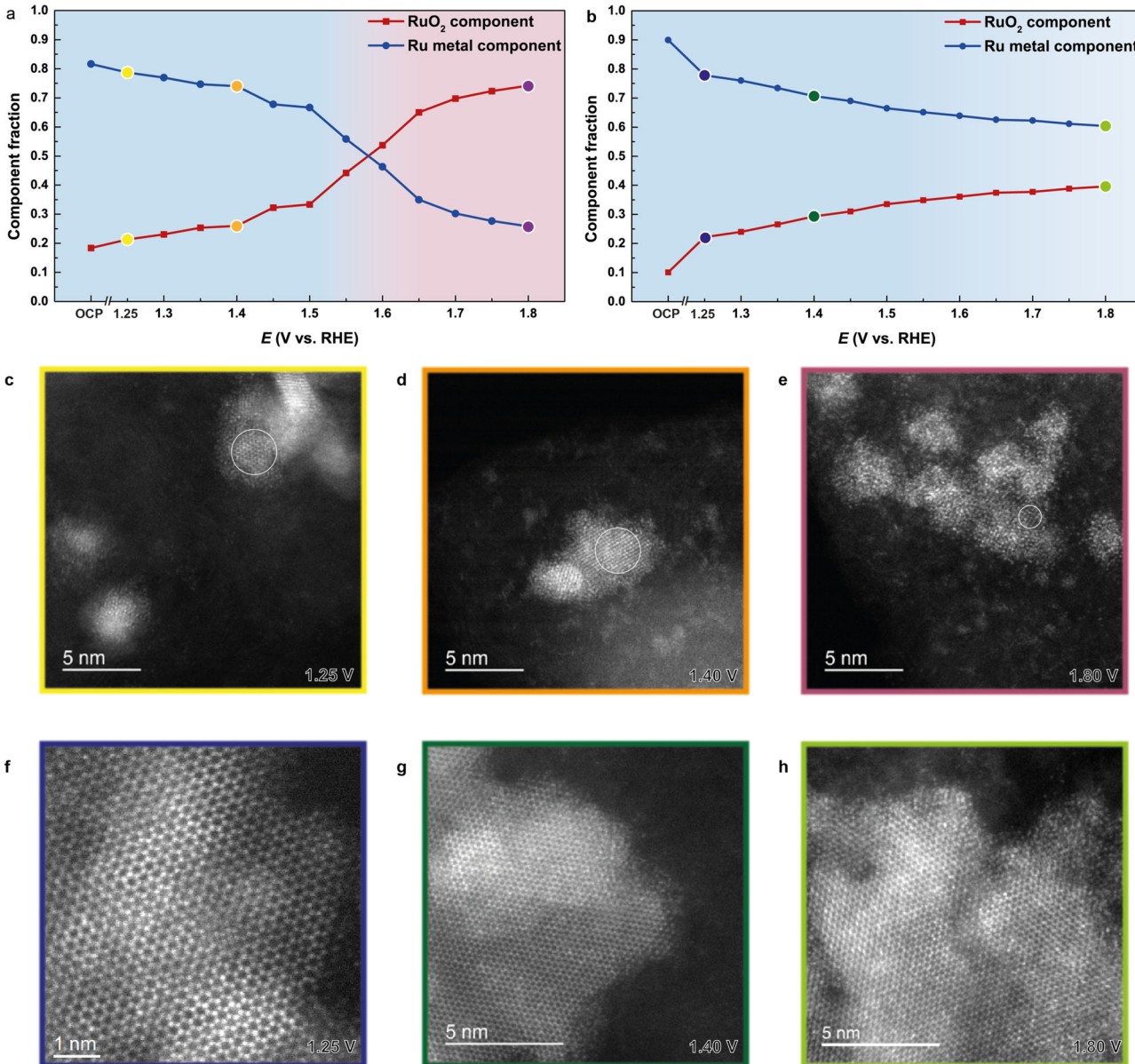

**Fig. 3 Potential-dependent local and surface structural changes.** Linear composition fitting (LCF) results using two components, Ru metal and $RuO_2$ in the X-ray absorption near-edge structure (XANES) spectra of **a RuIr-NS** and **b RuIr-NC** collected by operando tests. The coloured circles coincide with the colour outer boxes in (**c–h**). The background colours are used for guiding eyes. HAADF-STEM images showing the morphology changes in **c–e RuIr-NS** and **f–h RuIr-NC** under 1.25 V (**c**, **f**), 1.40 V (**d**, **g**) and 1.80 V (**e**, **h**). The circles in (**a–c**) show the crystalline part of the **RuIr-NS**.

was 46 mV which was lower than that of **RuIr-NS** (60 mV), Ir NPs (61 mV) and Ru NPs (98 mV), and comparable to that of commercial Pt/C (42 mV) (Fig. 4a). Ru or Ir NPs shows a Tafel slope of 81.0 or 42.8 mV/dec (Fig. 4b), respectively, which suggests a Volmer-Heyrovsky mechanism (Volmer: $H^+ + e^- \rightarrow H_{ad}$, 120 mV/dec, Heyrovsky: $H_{ad} + H_2O + e^- \rightarrow H_2 + OH^-$, 40 mV/dec in the theoretical calculation)[32,33], both are suffered from electron transfer. **RuIr-NS** has a Tafel slope of 38.3 mV/dec also showing a dominant Heyrovsky step. In contrast, **RuIr-NC** showed the smallest Tafel slope of 32.0 mV/dec among the synthesized catalysts. This value was close to that of commercial Pt/C (30.5 mV/dec) and suggests the efficient Tafel step ($H_{ad} + H_{ad} \rightarrow H_2$, 30 mV/dec in the theoretical calculation) (Fig. 4b). The RuIr catalysts show higher activity than monometallic catalysts, which suggests an alloying effect on the enhancement of HER activity. Moreover, theoretical calculation suggests that the 2D Ru(0001) sheets have a less negative Gibbs free energy change for hydrogen

and thus higher HER activity than their powder counterpart[34]. This suggests that the alloying effect and unique morphology might lead to high HER activity of **RuIr-NC**.

The good performance of **RuIr-NC** for both OER and HER allowed us to build a low-cost water electrolyzer in acid (named **RuIr-NC||RuIr-NC**). The voltage required to reach $10\ \text{mA cm}^{-2}_{geo}$ is as low as 1.485 V, which outperformed the combination of commercial catalysts $IrO_2||Pt/C$ (Fig. 4c and Supplementary Movies 2). Significantly, the cell is very stable at $10\ \text{mA cm}^{-2}_{geo}$ for over 120 h of operation (<5% increase in overpotential, Fig. 4d). From these results, **RuIr-NC** is the most efficient bifunctional catalyst for water-splitting outperforming all known acidic electrolyzers to date (Supplementary Table S5).

In summary, we reported Ru–Ir nanocatalyst with only 6 at.% Ir as a highly efficient overall water-splitting catalyst in the acidic solution. The obtained **RuIr-NC** are coral-like architecture consisting of highly crystalline 3-nm nanosheets with hcp solid-

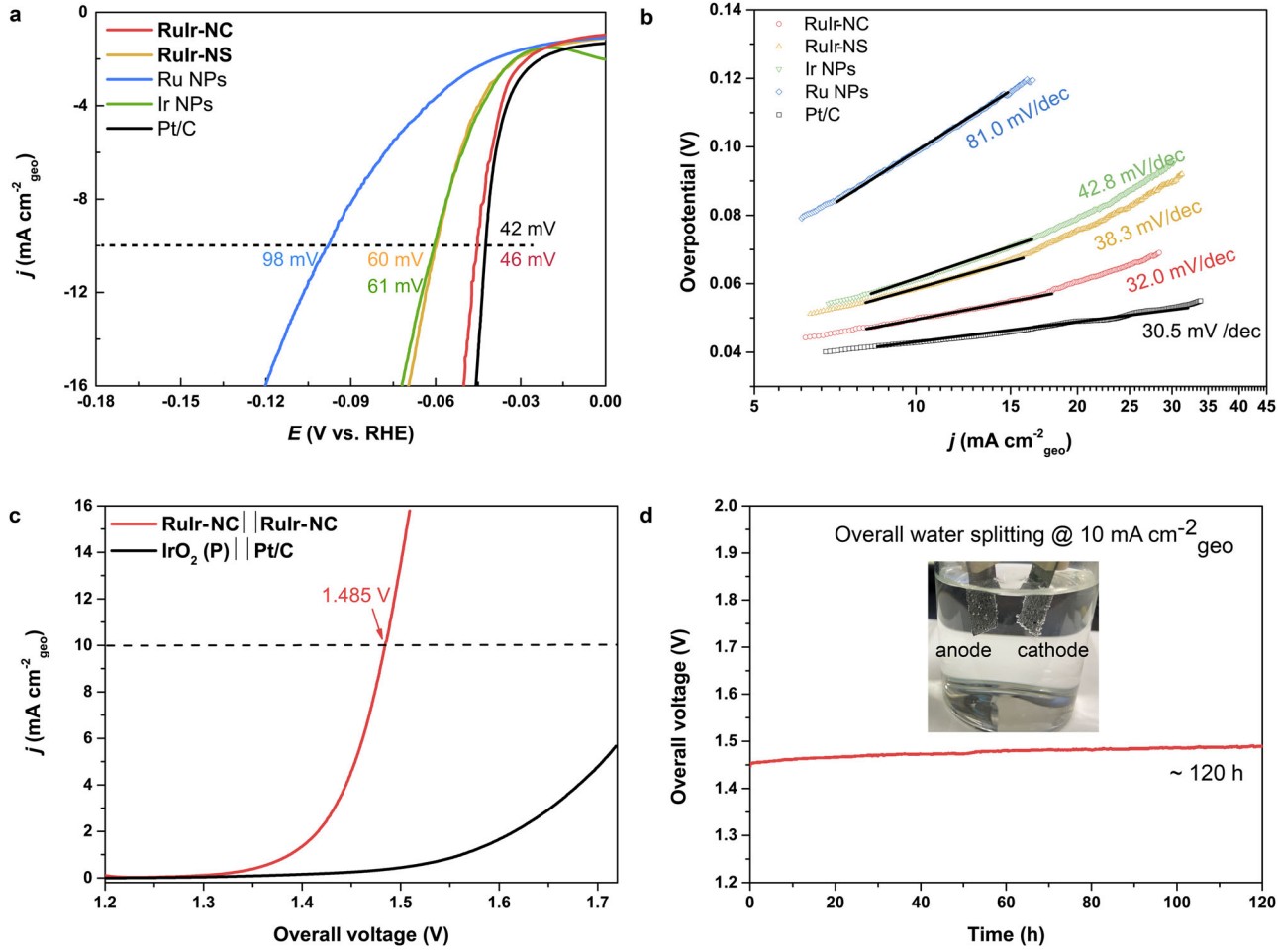

**Fig. 4 Hydrogen evolution reaction (HER) and overall water-splitting performance. a** HER polarization curves. The values are the overpotential to reach 10 mA cm$^{-2}_{\text{geo}}$. **b** Tafel plots. **c** Polarization curves of the two-electrode overall water-splitting cells (red: **RuIr-NC** as both electrodes, black: IrO$_2$(P) and Pt/C as OER and HER benchmarks, respectively). **d** Chronopotentiometric curves of **RuIr-NC ‖ RuIr-NC** at a current density of 10 mA cm$^{-2}_{\text{geo}}$. The inset shows the two-electrode configuration with bubbles on both electrodes.

solution structure. **RuIr-NC** is the most efficient OER catalyst in terms of geometric activity, mass activity and specific activity and shows remarkably high stability compared to other OER catalysts in acid. The overall water-splitting cell consisting of **RuIr-NC** also outperform others by realizing 10 mA cm$^{-2}_{\text{geo}}$ at 1.485 V for 120 h without noticeable degradation. The control experiments on ordinary spherical Ru–Ir catalysts and spectroscopy studies revealed that the unique nanosheets with extended {0001} facets contribute to the enhanced stability and activity. Although a dynamic structure change of the OER catalyst occurs with increasing potential, our result demonstrates that finely tuning the structure of original metal particles is a promising way for designing durable OER catalysts in acid.

## Methods

**Synthesis of RuIr-NC**. First, a solution of PVP (555 mg) dissolved in TEG (150 ml) was heated to 220 °C. Then, the mixture of RuCl$_3$·$n$H$_2$O ($n \approx 3$, 480.7 mg, 1.84 mmol) and H$_2$IrCl$_6$ (0.16 mmol) aqueous solution (15 ml) was added dropwise to the preheated solution. The temperature of the TEG solution was maintained at 220 °C while adding the precursors. After cooling to RT, the precipitate powder was separated from the resulting black solution by acetone follow by washing and vacuum dried process. Syntheses of other NPs are in Supporting information.

**General characterization**. Synchrotron XRD was measured at the BL02B2 beamline, SPring-8, Japan. The Rietveld refinement was performed using Pearson VII function[35,36] with TOPAS3 software developed by Bruker AXS GmbH. The atomic percentages were determined by XRF (ZSX Primus IV, Rigaku, Japan).

Bright-field TEM images were taken using an HT7700 microscope (Hitachi, Japan) operated at 100 kV. HAADF-STEM images and EDS were recorded on a JEM-ARM200F STEM instrument (JEOL, Japan) with an aberration corrector operated at 200 kV. ICP-MS was conducted on an ICPE-9000 (Shimazu, Japan). Lab XPS was performed on a Shimadzu ESCA-3400 X-ray photoelectron spectrometer (Shimazu, Japan). The HAXPES experiments were performed with a photon energy of 5.95 keV at the National Institute for Materials Science contract undulator beamline BL15XU at SPring-8, Japan. The binding energies were calibrated with respect to the Fermi edge of the Au reference. Both XPS and HAXPES spectra were calibrated by the C 1$s$ peak at 284.5 eV.

**HAADF-STEM 3D tomography and Ru/Ir ratio quantitative analyses**. The HAADF-STEM 3D tomography was reconstructed by a tilt series of HAADF-STEM images for the individual **RuIr-NC** acquired using the JEM-ARM200F at 120 kV. Tomographic reconstruction was performed by the discrete algebraic reconstruction technique (DART)[37,38] developed by Aarle et al.[39]. To determine the Ir positions and thickness, atomic column intensities were quantitatively analysed using the StatSTEM library developed by Backer et al.[40]. The single Ru and Ir atom intensity was carefully estimated based on the multislice method[41].

**Operando/ex situ XAFS**. XAFS experiments of the Ru $K$-edge and Ir $L_3$-edge were performed using a Si(311) monochromator crystal at the BL01B1 beamline at SPring-8, in Japan. All XAFS experiments were recorded in transmission mode at RT. The operando experiments were conducted with a home-made electrochemical cell (Supplementary Fig. 24). We collected the XAFS data at the OCP first and then performed quick XAFS (QXAFS) analysis during the polarization scan up to 1.8 V. To capture the change in the surface state of the catalysts, the scan rate was set as 0.05 mV/s, which is two magnitudes slower than that used in the lab experiments. For ex situ single measurements, the as-prepared materials were diluted with BN powder to make a pellet with a diameter of 7.0 or 10 mm.

XAFS analysis was performed using the Athena and Artemis program (version Demeter 0.9.26)[42]. PCA and target transformation (TT) were used to determine the number and type of principal components in the sets of 13 XANES spectra (Supplementary Figs. 20–23)[43]. Based on PCA, TT analyses and Roubaix diagram of Ru[44], Ru and RuO2 are the two components. The LCF was performed in the normalized XANES spectrum[43].

**Electrochemical measurements**. All the electrochemical measurements were conducted at RT and repeated five times to ensure reproducibility. To avoid the reaction of Ir with ClO⁻ ions in OER region[19], we used 0.05 M H2SO4 for water splitting. For the Cu UPD tests, 0.5 M H2SO4 containing 5 mM CuSO4 were used[45]. The OER and HER were performed with an RDE ($\varphi$ 5.00 mm, 0.196 cm$^2$) coupled with Ag/AgCl (in 3M NaCl) reference electrode and Pt wire (for OER) or graphite rod (for HER) counter electrode using a potentiostat (CHI 760e, USA). The catalyst inks were prepared by loading the as-prepared nanoparticles onto carbon (Vulcan XC-72R) with a metal weight percentage of approximating 20% (determined by elemental analysis). Then, the carbon-loaded catalysts (5 mg in total mass) were dispersed in 1 ml of a mixture solution of isopropanol (0.600 ml), water (0.300 ml) and 5 wt% Nafion (0.100 ml). Then, the catalyst ink (10 μl) was dropped onto the surface of the working electrodes (0.05 mg$_{metal}$ cm$^{-2}$) and dried under air overnight. For comparison, commercial Pt/C catalyst inks were prepared by following the same process.

For OER and HER, first, the electrodes were activated in an Ar-saturated 0.05 M (or 0.5 M) H2SO4 between 0.05 and 0.95 V at 500 mV s$^{-1}$ for several hundred CVs. Next, the electrodes were immersed at 0.05 V$_{RHE}$ (for HER) or 1.200 V$_{RHE}$ (for OER) for 6 min, followed by four LSV scans. The RDE was rotated at 1600 rpm with a scan rate of 5 mV s$^{-1}$. The CP experiment was carried out at a current density of 1 or 10 mA cm$^{-2}$. All OER polarization curves were background corrected to make the current before OER zero. The ohmic losses are corrected (with a percentage of 85%) based on the resistance obtained from electrochemical impedance spectroscopy (Supplementary Fig. 12). The overall water-splitting experiments were conducted in a home-made two-electrode cell with **RuIr-NC** as both the anode and cathode. The catalysts were loaded on carbon paper (an area of ca. 1*2 cm$^2$) with a loading amount of 0.15 mg cm$^{-2}$. For the benchmark commercial catalysts, highly conductive IrO2 (Premetek Co.) and Pt/C were employed as the cathodic and anodic catalysts, respectively. The long-term durability was evaluated using constant current electrolysis at 10 mA cm$^{-2}_{geo}$. For the Faradaic efficiency test of **RuIr-NC**, we monitored the generated oxygen amount by gas chromatography (GC) after fixing the electrode at each current density for 30 min. An H-cell and Nafion 117 film were used to separate the anodic and cathodic chamber.

## Data availability

All data are available in the main text or the supplementary materials, or from the corresponding author upon reasonable request.

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

## Acknowledgements

We acknowledge the support from JST ACCEL program Grant Number JPMJAC1501 and Grant-in-Aid for Specially Promoted Research 20H05623. STEM observations were performed as a part of a program conducted by the Advanced Characterization Nanotechnology Platform sponsored by the Ministry of Education, Culture, Sports, Science and Technology (MEXT) of the Japanese government. Synchrotron XRD measurements were performed at SPring-8 under proposal No. 2018A1215. The operando and ex situ XAFS experiment measurements were performed at SPring-8 under proposal Nos. 2018A1427, 2018B1353 and 2019A1123. The HAXPES measurements were performed at SPring-8 under proposal No. 2016B4910 and 2017A4910.

## Author contributions

D.W., K.K. and H. Kitagawa conceived the idea and designed the research. D.W. performed the synthesis, general characterization, and electrochemical tests. D.W., K.K., S.Y. and T.I. participated in the XAFS experiments and data analysis. T.Y., T.T. and S.M. conducted HAADF-STEM measurements and data analyses. Y.C., O. Seo, J.K., C.S., S.H. and O. Sakata performed the HAXPES measurements. S.K. and Y.K. contributed to the synchrotron XRD measurements. D.W., K.K., H. Kobayashi and H. Kitagawa discussed the results and wrote the paper. All the authors discussed and commented on the paper.

## Competing interests

The authors (D.W., K.K. and H. Kitagawa) have a patent application: anisotropic nanostructure, production method therefor, and catalyst, WO 2019121744A1. D.W., K.K. and H. Kitagawa are related to this research. All other authors declare no competing interests.
