## [Peer Review File · Nature Communications]

REVIEWER COMMENTS

Reviewer #1 (Remarks to the Author):

This manuscript presented a unique coral-like RuIr architecture for highly active and stable water splitting in acid media. RuIr NC catalysts exhibited an overpotential of 170 mV at 10 mA/cm² for oxygen evolution reaction in acid. Moreover, using both RuIr NC as OER and HER catalysts, the water splitting system delivered a current density of 10 mA/cm² at 1.485 V and a long stability of 120 hours. The authors claimed that the coral-like nanostructure with exposed {0001} facets is the exclusive factor that contributes the enhanced activity and stability. Overall, the performance is impressive and the work is interesting. However, there are still several issues. Reliable experimental data still lacks to support their conclusions. Therefore, I don't think this manuscript is qualified enough to publish in Nature Communications at the current stage.

1. The authors claimed that EDS maps (Fig. 1g-i) demonstrate the homogeneous distribution of Ir and Ru atoms in RuIr NC. Firstly, the corresponding EDS spectrum associated with the EDS mapping should be shown. Secondly, the mapping resolution in Figure 1g-i is not high enough to prove the homogeneous distribution of Ir and Ru. Finally, the authors mentioned that the atomic ration of Ru and Ir is 0.96:0.04. If that is the case, what would the final structure of RuIr NC be? Ru₉₆Ir₄? More evidence regarding how the RuIr is structured should be given.

2. In the experimental section, the authors wrote that "all polarization curves were background corrected to make the current before OER zero". It should be explained where the background current comes from and how this process is done.

3. From the XANES and EXAFS analysis, the authors claimed that RuIr NC and RuIr NS have the similar electronic structures of Ru and Ir (Supplementary Fig. 15a, b and Supplementary Fig. 16). It is not surprising that HAXPES showed exactly overlapped Ru3p and Ir4f core level spectra for RuIr NC and RuIr NS. Ex situ studies could not reveal the real electronic structures of catalysts during reactions.

4. Supplementary Fig.16 showed oxidation states of Ir is less than 4+, which is contradictory with the published paper Nature Catalysis 1, 841-851 (2018). The IrNiOx catalysts in Nature Catalysis 1, 841-851 (2018) has high activity and high oxidation states.

5. Detailed analysis and evidence should be added to prove that the coral-like nanostructure with exposed {0001} facets contributes the enhanced activity and stability.

6. HAADF-STEM images showed mixed crystalline and amorphous structures for RuIr NS, while the hexagonal lattice of hcp(0001) for RuIr NC. Why RuIr NS easily form separated crystalline and amorphous structure? What are the amorphous and crystalline phases?

7. Figure 2c showed the stability of the RuIr NC at 1 mA/cm². How about the stability at higher current densities, such as 10 mA/cm².

8. The authors should also explain the hydrogen evolution activity improvement.

Reviewer #2 (Remarks to the Author):

Manuscript NCOMMS-20-23182 reports an interesting study about Ru-Ir electrocatalysts for the OER (and HER) in acid electrolyte. This reaction is of great interest for industry and academia mostly because hydrogen is expected to have a pivotal role for the transformation of the energy system. In particular, this manuscript reports that Ru-Ir particles, with only 6 at.% Ir and displaying a coral-like structure, display high OER activity and durability in acid electrolyte. The high OER performance of these particles has been ascribed to the presence of {0001} facets. This result is very interesting, and the presence of such facets in the Ru-Ir NC has been observed by XRD, the reason for the enhanced catalytic performance of such phase is not analysed. What is so especial about this crystalline phase? Theoretical studies will be most welcome to understand the actual role of such phase.

Although characterization results of the fresh samples by TEM and XRD are convincing, several features are not fully convincing. XRD reveals the presence of a diffraction peak for the 100 phase in the Ru-Ir NC; this peak is totally missing in the diffractogram for Ru-Ir NS sample. This is strange

since if this peak is not from a forbidden diffraction it should be observed in the diffractograms. The preferential orientation would result in higher intensity of such peak, but only if the peak can be observed. A magnification of the 2 theta region between 0- and 20° should be shown. The position of the Bragg peaks for the RuIr structures in Figures 4 and 5 should be shown.

It is not clear to me how the ECSA (or EASA) are calculated, is it from COad stripping or from Cu UPD? Determining Ru or Ir's surface area from COad stripping is not straightforward since the CO-M stoichiometry is not known (polycarbonyl species are usually formed). Therefore, the 0.42 mC/cm² relation employed for determining Pt surface area does not apply to other metals since it depends on the number of atoms per cm² on the polycrystalline surface.

The activity reported for Ru-Ir NC is rather high, higher in fact than that of the reference materials used in this work. However, the choice of such reference materials can be questionable since the actual performance of IrOx and RuOx nanosized particles depends on features such as particle size. In addition, the activity is similar to that reported for state-of-the-art Ru based catalysts, as shown in Table S4. As for the mass activity (or specific activity), the real figure of merit would be the activity per gram of metal (Ru+Ir) rather than the catalysts loading most of the catalysts reported in the literature are oxides, therefore containing a lower amounts of Ru or Ir than the one used in this work. Cyclic voltammograms up to 1.8 V should be shown, showing several cycles. This potential has been chosen by the authors to conduct their stability study. How is double layer corrected? Also, the raw voltammograms, without double layer correction should be shown. If potentials have been iR corrected, it should be indicated in the figures. What is the resistance obtained? Usually, OER is studied under O₂ saturated electrodes. What is the area of the electrodes used in the home-made electrolyser?

For the understanding of the durability, the authors have analysed the dissolution of Ru and Ir during repeated polarization. What is the actual program used? The experimental section states that long-term stability has been evaluated at 10 mA/cm², but in some cases the durability tests have been conducted at 1 mA/cm² (see Fig 2c), this current density is not representative for the performance of an electrolyser.

After durability tests, it is concluded that Ru-Ir NC has very high stability because had only 25% and 15% of Ru and Ir loss, respectively, even after five scans. Actually, this is a severe loss of material for only 5 cycles. In addition, this test fails short to assess material's durability since in many cases, especially with Ru based catalysts, activity drop faster with the increasing number of scans.

The bulk composition of both Ru-Ir particles seems to be similar, but what about the surface composition, which is actually more relevant for the catalytic performance. What is the Ru/Ir surface atomic ratios in both samples? What are the oxidation states of surface Ru and Ir in both samples? XAS analysis identifies RuO₂ in the sample subjected to 1.8 V. Have the authors observed Ru (and Ir) atoms in upper oxidation states? Ru tends to form perruthenates at such high potentials. The formation of oxides, and their actual nature, is very relevant for the activity of Ru and Ir based catalysts for the OER. Is indeed the development of an amorphous oxide layer observed by STEM related to the OER activity of the samples?

Subjecting Ir and Ru to high potentials in acid electrolyte would most like lead to material's corrosion, which result in strong positive currents. What fraction of the current reported in this work accounts to the OER and what fraction accounts to materials corrosion?

The conclusion that a cheaper electrolyser has been constructed is not sustained. What is the price of each electrolyser? Moreover, catalysts are not the most expensive components of PEM water electrolysers.

Reviewer #3 (Remarks to the Author):

In this manuscript, the authors report the synthesis of RuIr nano-coral structures and their electrocatalytic performance for OER, HER and overall water splitting. They have carried out very comprehensive structural, compositional and electrochemical characterization, and demonstrated excellent performance of the RuIr-NC with respect to other control catalysts. However, there are many

incorrect statement and some improper practice which make the data presented in this manuscript not fully convincing. For these reasons, the reviewer does not recommend the Editor to accept the present manuscript for publication in Nat. Commun. The reviewer's specific comments are summarized below:

1. There are some statements that are incorrect. For instance:

Abstract: "predominately because of the high cost and low performance of catalysts promoting the oxygen evolution reaction". This is not the major reason that can explain the small share of electrolysis-derived hydrogen in the market. In fact, the cost of catalysts only accounts for a small portion of the system costs of electrolyzers (see e.g. <https://www.nrel.gov/docs/fy19osti/72740.pdf>).
Line 48, page 2: the authors claim that "the design principles for their use in acid have not been well developed". In fact, there are already several review articles available about OER catalysts in acids (e.g. Adv. Energy Mater. 2016, 1601275), where how to design and optimize catalysts to achieve high OER performance is pointed out.

Lines 136-137: This statement is not correct. The mass activity, by definition, is derived simply by normalizing the catalytic current to the catalyst mass. It has nothing to do with EASA. The specific activity is closely linked to EASA.

Lines 241-243: Bifunctional catalysts have been investigated extensively in recent years. Many noble metal based OER catalysts also show reasonably good HER performance.

2. Line 37, page 2: what does "are ready at ..." mean? Practical PEM water electrolysis operates at a current density of 1 A/cm² or above, are the authors sure that only millivolts are needed in this case?

3. Line 65, page 3: "practical current density" appears several times in the manuscript. Can the authors specify what they think is the practical current density? For PEM water electrolysis, a practical current density should be 1 A/cm² or even higher. For solar water splitting, the current density is usually in the range of 10 – 20 mA/cm².

4. The authors mention several times "the known systems". This is confusing. According to the work reported here, they are not working at a "system"(electrolyzer) level.

5. Page 4: discussion about XRD data – parentheses should be used when mentioning the indexed facets (lines 96-97)

6. The authors claim that there are 4 at% Ir in the RuIr-NC according to the EDS result. What spectral lines (L or M) did they use for the quantitative analysis? An EDS spectrum should be given. Moreover, according to the signal intensity of Figure 1h, it does not look that the atomic percentage of Ir is only 4%. Did the authors manipulate the intensity manually in the software?

7. The authors claim that (0001) crystal face is exposed. However, from Figure 1c, there are obviously at least two different regions where the atoms are arranged in different ways. How do the authors explain this?

8. Figure 2a: the data showing here are questionable. Where couldn't Ru NPs reach 10 mA/cm²? It seems that the authors swept the potential positively, which is not a proper practice for OER assessment because in this case the oxidation from redox will appear leading possibly to an overestimation of the activity. The potential should be swept negatively in order to avoid the overestimation. Furthermore, the LSV of Ru NPs is deformed, which may result from the over iR-correction. The degree of iR-correct is not given in the manuscript.

9. The RuIr-NS and RuIr-NC were synthesized using the same procedure. Then under which conditions can one obtain RuIr-NC and which other conditions to obtain RuIr-NS? Why? This should be discussed in more detail in the manuscript.

10. The authors failed to specify the loading mass of each catalyst during the HER or OER test, which influences the geometric activity.

11. Lines 140-142: The reviewer does not agree with this statement. For Cu UPD, it is likely that Cu cannot be deposited on every catalytically active sites because of the mass transfer limitation. In this case, the real catalytically active surface is in fact underestimated, while the specific activity is overestimated. The thus-obtained specific activity is the maximal possible one. It seems that the authors mix up the mass activity and specific activity. The fact that "the total mass content of metals" is considered actually verifies that what the authors state is wrong.

12. The authors assessed the stability at a low current density of 1 mA/cm² in such a way they've observed a long catalytic stability of 122 h. Why didn't they perform the test at 10 mA/cm²? For PEC water splitting, 10 mA/cm² is a sensible current density; while for electrocatalytic water splitting,

catalysts should be able to sustain at much higher current densities for a long term.

13. In Table S1, the authors overlooked some high-performance Ir based OER catalysts developed recently. They are recommended to carefully check recent publications and included the latest advances for comparison.

14. Lines 170-176: The RuIr-NC lost its Ir and Ru by 15% and 25%, respectively, only after a few polarization scans, which are significant actually. This implies that at a high current density (potential) RuIr-NC is likely unstable. That was probably why the stability was only assessed at 1 mA/cm².

15. Lines 190-192: The authors should briefly mention how the isotropic Ru-Ir catalyst was prepared in the main text.

16. Fig. S18: From the Tem micrograph, the size of RuIr-L is significantly increased. This likely explains why RuIr-L has poorer performance. This is contradict with the authors' claim that "the crystal size is not a determined reason". What is the EASA of RuIr-L?

17. In Figure 4d inset and the movies S1&S2, the authors emerged the electrode holders into the acidic solution for testing, which is perhaps not proper practice. What is the material of electrical contact? If it is Pt, it may contribute to the electrocatalytic process as the solution can easily access to the Pt sheet. Corrosion might also happen in this case. The electrical contact should be protected from the attack of corrosive acidic electrolyte (using either epoxy resin or simply lifting the holder to a place far away from the electrolyte surface).

18. Fig. S8: Histograms showing the size distribution of the samples should be given as insets.

19. How was the catalyst ink prepared? From Fig. S27, it seems that the RuIr-NC was loaded on carbon support?

20. Lines 248-249: Given the Tafel slope, both Ru and Ir NPs should follow a Volmer-Heyrovsky mechanism during the HER. The Volmer mechanism works when the Tafel slope exceeds 120 mV/dec. BTW, it should be "Heyrovsky", rather than "Heyrosky".

21. Line 259: I would suggest the authors use "practical" with caution. With these "practical" catalysts, PEM water electrolysis can be accomplished at 500 mA/cm² under ca. 1.5 V (see e.g. J. Electrochem. Soc. 2018, 165, F305), much better than the RuIr-NC bifunctional catalysts.

22. The authors are recommended to improve their English. There are a number of grammatical errors, typos and inappropriate usage of English. Just to name a few:

Line 53, page 3: what does "heave Ir doping" mean? Should it be "heavy"?

Line 69, page 3: "operating" should be inserted after "keeps"

Lines 95-96, page 4: instead of "...size...were...", it should be "...sizes...are..."

Line 103: what do the squares mean?

Line 139: it should be "...catalysts... take...", rather than "takes"

Line 142: According to Fig. S12d, the specific activity should be 4.9 mA/cm², but not 4.9 A/cm²

Line 170: "valid" should be corrected to "validate"

And many others...

Response Letter

Reviewer #1

This manuscript presented a unique coral-like Rulr architecture for highly active and stable water splitting in acid media. Rulr NC catalysts exhibited an overpotential of 170 mV at 10 mA/cm² for oxygen evolution reaction in acid. Moreover, using both Rulr NC as OER and HER catalysts, the water splitting system delivered a current density of 10 mA/cm² at 1.485 V and a long stability of 120 hours. The authors claimed that the coral-like nanostructure with exposed {0001} facets is the exclusive factor that contributes the enhanced activity and stability. Overall, the performance is impressive and the work is interesting. However, there are still several issues. Reliable experimental data still lacks to support their conclusions. Therefore, I don't think this manuscript is qualified enough to publish in Nature Communications at the current stage.

Q1. The authors claimed that EDS maps (Fig. 1g–i) demonstrate the homogeneous distribution of Ir and Ru atoms in Rulr NC. Firstly, the corresponding EDS spectrum associated with the EDS mapping should be shown. Secondly, the mapping resolution in Figure 1g-i is not high enough to prove the homogeneous distribution of Ir and Ru. Finally, the authors mentioned that the atomic ration of Ru and Ir is 0.96:0.04. If that is the case, what would the final structure of Rulr NC be? Ru₉₆Ir₄? More evidence regarding how the Rulr is structured should be given.

Answer: We appreciate the reviewer's comments and suggestions.

The corresponding EDS spectrum is added in the Supplementary Fig. 10 (see below).

To show the whole area of a nanocoral, we adopted the magnification in Fig. 1g-i. Although we tried to obtain EDS maps with higher magnification, it was difficult to clearly distinguish the real Ir signal in a nanocoral from the background due to the very low concentration of Ir. Therefore, to prove the homogenous and random distribution of Ir in Ru lattice, we provided the quantitative analysis of Ir atom number in a single column from

HAADF image because HAADF is sensitive to the atomic numbers (Z-contrast) (Fig. 1 e and f).

According to your suggestion, the final structure of **RuIr-NC** was observed by HAADF-STEM and EDX analyses and the results were added in Supplementary Fig. 32 and Table 4. The **RuIr-NC** maintained the coral-like structure and its original size distribution after OER. The Ir concentration of **RuIr-NC** after OER (Ru: Ir = 92.8: 7.8) is slightly higher than the as-prepared sample. However, the percentage of Ir in **RuIr-NS** much more increased at 1.8 V, which implied that the dissolution rate of Ru in **RuIr-NS** is much higher than that of **RuIr-NC**.

Revision 1: Page 9, Line 234, “The Ru/Ir compositions of the catalysts during OER were measured by EDX (Supplementary Table 4). Before 1.4 V, the Ru/Ir compositions in **RuIr-NS** and **RuIr-NC** samples were not changed so much. However, the percentage of Ir in **RuIr-NS** much more increased at 1.8 V. In contrast, the percentage of Ru and Ir in **RuIr-NC** did not obviously change even at 1.8 V. This implied that the dissolution rate of Ru in **RuIr-NS** is much higher than that of **RuIr-NC**.”

Page 10, Line 248, “the coral-like shape showing the hexagonal lattice of the hcp (0001) plane is mainly observed up to 1.80 V and size distribution of the catalyst is maintained (Fig. 3f–h and Supplementary Fig. 32)”

In the Supplementary Information, we added Fig. 10, Fig.32 and Table 4, respectively.

Supplementary Fig.10. EDS spectrum of the particle shown in Fig.1b. Clear Ru and Ir peaks were shown. Other signals were system peaks from the STEM and its grid.

Supplementary Fig. 32 STEM images of **RuIr-NC** after polarization scans to different potentials. **a** and **d**, 1.2 V. **b** and **e**, 1.40 V. **c** and **f**, 1.80 V. The scale bars for **a-c** and **d-f** are 500 and 50 nm, respectively. The size distributions at 1.2 V, 1.4 V and 1.8 V are 53 ± 9.0 , 51 ± 15 , and 54 ± 13 nm, respectively. The nanocoral maintained its morphology and size up to 1.80 V.

Supplementary Table 4 The atomic percentage (%) of Ru and Ir in the as-prepared RuIr catalysts and RuIr catalysts during OER.

	As-prepared	EDX @ JEOL ARM-200 kV			XRF	XPS
		1.2 V	1.4 V	1.8 V	As-prepared	As-prepared
RuIr-NS	95.9 / 4.1	94.5 / 5.4	95.2 / 4.8	83.8 / 16.2	93.5 / 6.5	92.0 / 8.0
RuIr-NC	96.0 / 4.0	96.0 / 4.0	93.8 / 6.2	92.8 / 7.8	93.6 / 6.4	93.5 / 6.5

Q2. In the experimental section, the authors wrote that "all polarization curves were background corrected to make the current before OER zero". It should be explained where the background current comes from and how this process is done.

Answer: The background current comes from the capacitance current and the oxidation of carbon and metals. We directly subtracting the current at 1.2 V just before OER in an LSV scan and the process is presented as Supplementary Fig.12 a, b. With the background-correction, the current before the onset of OER is almost zero.

Revision 2: Page 5, Line 130, “All the LSV curves are background current and ohmic corrected and the currents contributed from the bare RDE and carbon/Nafion mixture at higher potentials are negligible (Supplementary Fig. 12)”

In the Supplementary Information, we added Fig. 12 a, b

Supplementary Fig. 12 Ohmic and background corrections of the OER activity of **RuIr-NC**. **a**, polarization curve of **RuIr-NC** (“Raw data”, solid red line, shown in the range of 1.15 – 1.35 V) was corrected by subtracting the background current density. The current density at 1.2 V, 0.25 mA cm^{-2} , is considered as the background current density which comes from the capacitance current and oxidation of metal and carbon before OER. With the background-correction, the current is almost zero before the onset of OER (dashed blue line). **b**, the ohmic-corrected OER current (dashed blue line) is then corrected with the measured resistance ($\approx 38.3 \Omega$) to yield the final electrode OER activity (solid black line).

Q3. From the XANES and EXAFS analysis, the authors claimed that RuIr NC and RuIr NS have the similar electronic structures of Ru and Ir (Supplementary Fig. 15a, b and Supplementary Fig. 16). It is not surprising that HAXPES showed exactly overlapped Ru3p and Ir4f core level spectra for RuIr NC and RuIr NS. Ex situ studies could not reveal the real electronic structures of catalysts during reactions.

Answer: Thanks for your comments. From Supplementary Fig. 15a, b and 16 (now Fig. 18 a, b and 20), what we want to say is that the electronic structures of **RuIr-NC** and **RuIr-NS** are similar and both samples have similar Ir sites before the electrochemical test. The electronic structures or valence states of catalysts during the reaction is shown in Fig. 3 and Supplementary Fig. 25 – 29.

Q4. Supplementary Fig.16 showed oxidation states of Ir is less than 4+, which is contradictory with the published paper Nature Catalysis 1, 841-851 (2018). The IrNiOx catalysts in Nature Catalysis 1, 841-851 (2018) has high activity and high oxidation states.

Answer: Thank you for your question. Supplementary Fig. 16 (Now Fig. 20) shows the initial state of Ir in metallic RuIr catalysts before reaction. During OER, the gradual oxidation of Ir can be verified from the XRD (Supplementary Fig. 30) and XPS (Supplementary Fig. 18) after OER. We believe that the active species with higher oxidation states will be obtained just before or during OER. This is consistent with the polished Nature Catalysis Paper.

Q5. Detailed analysis and evidence should be added to prove that the coral-like nanostructure with exposed {0001} facets contributes the enhanced activity and stability.

Answer: Thanks for your suggestions. It is known that the single-crystalline Ru(0001) facet is more resistant to the oxidization and dissolution than the other facets such as Ru(10 $\bar{1}$ 0) (Ref. 31, Lin, W. et al., *J. Phys. Chem B.* 104, 6040-6048 (2000), and Ref. 75, Özer E. et al., *ChemCatChem*, 9, 597-603 (2017)). The stability of (0001) facet can contribute to the high durability and activity of **RuIr-NC**, especially at high potentials. Following your suggestion, to further confirm that the exposed {0001} facets of **RuIr-NC** contribute to OER performance, we added the CV results of **RuIr-NC** and **RuIr-NS** in the Supplementary Fig. 34. Compared to **RuIr-NS**, the CV feature of **RuIr-NC** is more similar to that of Ru(0001) single-crystalline electrode obtained in Ar-saturated 0.05 M H₂SO₄ (Ref. 75). **RuIr-NC** shows two reduction peaks at around 0.22 V and 0.40 V and an

oxidation peak at around 0.60 V, even though those peaks broaden probably due to the nanosize and/or Ir doping effects. These CV results support our conclusion that **Rulr-NC** mainly has {0001}-terminated facets, which contributes the enhanced activity and stability.

Revision 3: Main text, Page 10, Line 257, “CV of **Rulr-NC** in Ar-saturated 0.05 M H₂SO₄ shows similar feature to that of single-crystalline Ru(0001) electrode (Supplementary Fig. 34), which is consistent with the STEM analyses and verifies that (0001) facet of hcp structure might protect Ru from corrosion”.

In the Supplementary Information, we added Fig. 34.

Supplementary Fig. 34 CVs of Rulr catalysts and single-crystalline Ru (0001) electrodes in Ar-saturated 0.05 M H₂SO₄. **a**, CVs of **Rulr-NC** and **Rulr-NS** recorded at a scan rate of 20 mV/s. **b**, CVs of single-crystalline Ru (0001) electrode reported in Ref. 75..

Q6. HAADF-STEM images showed mixed crystalline and amorphous structures for Rulr NS, while the hexagonal lattice of hcp(0001) for Rulr NC. Why Rulr NS easily form separated crystalline and amorphous structure? What are the amorphous and crystalline phases?

Answer: The amorphous phases are the (hydro)oxides formed on the surface of metal NPs during or just before OER. With increasing the potential, the oxidation of Ru or Ir will happen and the amorphous (hydro)oxides structures grow from the surface of metal NPs. The crystalline parts are the remaining metallic parts and surrounded by the amorphous part. We have to say that **Rulr-NC** also has amorphous parts as shown in the

Supplementary Fig. 33. There is a thinner (< 0.5 nm) amorphous part along with the $[0001]$ direction and thicker (ca. 5.0 nm) one along with the $[10\bar{1}0]$ direction. For the HAADF-STEM images captured from the $[0001]$ projection (shown in Fig. 3f-h in the main text), the thickness of amorphous oxides (< 0.5 nm) is too thin to be observed.

Supplementary Fig. 33 HAADF-STEM image view from the $[01\bar{1}0]$ direction of **RuIr-NC** at 1.8 V.

Revision 4: Main text, Page 10, Line 257, “CV of **RuIr-NC** in Ar-saturated 0.05 M H_2SO_4 shows similar feature to that of single-crystalline Ru(0001) electrode (Supplementary Fig. 34), which is consistent with the STEM analyses and verifies that (0001) facet of hcp structure might protect Ru from corrosion”.

Q7. Figure 2c showed the stability of the RuIr NC at 1 mA/cm². How about the stability at higher current densities, such as 10 mA/cm².

Answer: Following your suggestion, we performed the stability measurement at 10 mA/cm² and added the data in the Supplementary Fig.14 and Table 1. The **RuIr-NC** is stable for at least 40 h, which is better than the other catalysts list in Table 1.

Revision 5: The main text, Page 7, Line 164: “The **RuIr-NC** can sustain 40 h even under 10 mA cm⁻²_{geo} (Supplementary Fig. 17)”.

Supplementary Information, we added Fig. 17 and modified in Table 1.

Supplementary Fig. 17. CP curve of **RuIr-NC** at the OER current density of $10 \text{ mA cm}^{-2}_{\text{geo}}$. The **RuIr-NC** can sustain its activity for more than 40 h (with less than 5% increase of the potential).

Q8. The authors should also explain the hydrogen evolution activity improvement.

Answer: Thank you for pointing out this point. We explain the HER properties in terms of the alloying effect and shape effect as below.

Revision 6: Page 11, Line 277, we added “The RuIr catalysts shows higher activity than monometallic catalysts, which suggests an alloying effect on the enhancement of HER activity. Moreover, theoretical calculation suggests that the 2D Ru(0001) sheets have a less negative Gibbs free energy change for hydrogen and thus a higher HER activity than their powder counterpart³⁵. This suggests that alloy effect and unique morphology might lead to high HER activity of **RuIr-NC**.”

Reviewer #2

Q1. Manuscript NCOMMS-20-23182 reports an interesting study about Ru-Ir electrocatalysts for the OER (and HER) in acid electrolyte. This reaction is of great interest for industry and academia mostly because hydrogen is expected to have a pivotal role for the transformation of the energy system. In particular, this manuscript reports that Ru-Ir particles, with only 6 at.% Ir and displaying a coral-like structure, display high OER activity and durability in acid electrolyte. The high OER performance of these particles has been ascribed to the presence of {0001} facets. This result is very interesting, and the presence of such facets in the Ru-Ir NC has been observed by XRD, the reason for the enhanced catalytic performance of such phase is not analysed. What is so especial about this crystalline phase? Theoretical studies will be most welcome to understand the actual role of such phase.

Answer: Thank you very much for your nice suggestions.

For metal-based catalysts, the active species is the amorphous (hydro)oxides generated during (or just before) OER. From the LSV shown in Fig.2a, **RuIr-NC** and **RuIr-NS** have similar activity at the low potential range (below 1.4 V). This suggests that active metal oxides on these two RuIr catalysts might be similar. In the higher potential range, **RuIr-NC** shows higher activity than **RuIr-NS**. This is because the **RuIr-NC** greatly suppresses the severe dissolution compared to **RuIr-NS**. Ru(0001) single-crystalline electrode is known to be more resistant to the oxidization and dissolution than the other facets such as Ru(10 $\bar{1}$ 0) during electrochemistry (Ref. 31, Lin, W. et al., *J. Phys. Chem B*, 104, 6040-6048 (2000), and Ref. 75, Özer E. et al., *ChemCatChem*, 9, 597-603 (2017)). In the manuscript, we have already clarified that **RuIr-NC** has extended hcp(0001) plane, and inferred that the high activity of **RuIr-NC** at higher potentials mainly comes from the high stability of hcp(0001) facet which more resists to oxidation and protects the **RuIr-NC** from further corrosion.

We agree with the referee's suggestion that a theoretical study can give an insight into its mechanism. However, it is very difficult to construct suitable models of the amorphous structures on the catalysts surface, and we failed to obtain reliable theoretical

results. Therefore, to support our discussion, we performed further electrochemical experiments. Since electrochemical reactions happen at the very surface of solid catalysts, to directly investigate if (0001) facet is really exposed and affects the performance during the electrochemistry process, we performed CV of RuIr catalysts and compared their CV features with that of single-crystalline Ru(0001) electrode. Compared to **RuIr-NS**, the CV feature of **RuIr-NC** is more similar to that of Ru(0001) single-crystalline electrode (Ref. 75). **RuIr-NC** shows two reduction peaks at around 0.22 V and 0.40 V and one oxidation peak at around 0.60 V, even though those peaks broaden due to the nanosize and/or Ir doping effects. This CV result revealed that **RuIr-NC** with mainly {0001}-terminated surface would have enhanced stability and activity, especially at higher potentials. We added these results in both the main text and the Supplementary Information.

Revision 3: Main text, Page 10, Line 257, “CV of **RuIr-NC** in Ar-saturated 0.05 M H₂SO₄ was very similar to that of single-crystalline Ru(0001) electrode (Supplementary Fig. 34), which is consistent with the STEM analyses and verifies that (0001) facet of hcp structure might protect Ru from corrosion”.

Supplementary Information, we added Fig. 34.

Supplementary Fig. 34 CVs of RuIr catalysts and single-crystalline Ru (0001) electrodes in Ar-saturated 0.05 M H₂SO₄. **a**, CVs of **RuIr-NC** and **RuIr-NS** recorded at a scan rate of 20 mV/s. **b**, CVs of single-crystalline Ru (0001) electrode reported in Ref. 75.

Q2. Although characterization results of the fresh samples by TEM and XRD are convincing, several features are not fully convincing. XRD reveals the presence of a diffraction peak for the 100 phase in the Ru-Ir NC; this peak is totally missing in the diffractogram for Ru-Ir NS sample. This is strange since if this peak is not from a forbidden diffraction it should be observed in the diffractograms. The preferential orientation would result in higher intensity of such peak, but only if the peak can be observed. A magnification of the 2 theta region between 0- and 20° should be shown. The position of the Bragg peaks for the Rulr structures in Figures 4 and 5 should be shown.

Answer: The 100 phase [(10-10) phase] in **Rulr-NS** is not missing and it is located at the 2θ position of 14.28° . We added the positions of Bragg peaks in the Supplementary Fig. 4 and 5 (see Figures below). The magnification of the 2θ region from 0 to 20° is shown in Figure A below. In the range before 12° , there are no peaks except for the broaden signals from the glass capillary and PVP.

Figure A. XRD pattern in the range of 0-20°.

Revision 7: In the Supplementary Information, we added the position of Bragg peaks for the Rulr structures in Fig. 4 and 5.

Supplementary Fig. 4 XRD patterns and Rietveld refinement of **Rulr-NC**. The black circles are the experimental results. The red line is the calculated pattern. The bottom lines show the difference profile (grey) and the background item (light-blue). Crystal sizes obtained from different orientations are given in the inset presenting the extended (0001) plane in **Rulr-NC**. Inset table shows the Bragg positions.

Supplementary Fig. 5 XRD patterns and Rietveld refinement of **Rulr-NS**. The black circles are the experimental results. The red line is the calculated pattern. The bottom lines show the difference profile (grey) and the background item (light-blue). The table shows the Bragg positions.

Q3. It is not clear to me how the ECSA (or EASA) are calculated, is it from COad stripping or from Cu UPD? Determining Ru or Ir's surface area from COad stripping is not straightforward since the CO-M stoichiometry is not known (polycarbonyl species are usually formed). Therefore, the 0.42 mC/cm2 relation employed for determining Pt surface area does not apply to other metals since it depends on the number of atoms per cm2 on the polycrystalline surface.

Answer: Thank you very much for pointing out our mistakes. We determined the EASA by Cu UPD method. We corrected the experimental part in the supplementary section. To verify the accuracy of Cu UPD method, we also evaluate the surface area of **RuIr-NS** by the TEM based on a spherical geometry (Ref. 26, Lee, Y., et. al., *J. Phys. Chem. Lett.* **3**, 399-404 (2012)). The EASA values of **RuIr-NS** obtained by two methods are quite similar (114 vs. 111 m²/g, Supplementary Fig. 15c).

Revision 8: In the Supplementary Information, we corrected the method as “For the Cu UPD experiments⁴⁴, first, high-purity Ar was bubbled through the H₂SO₄ electrolyte (0.5 M) for at least 15 min to remove the O₂ in the electrolyte. After Ar saturation, the catalysts underwent electrochemical pre-treatment by potential cycling between 0.05 and 0.95 V for 200 cycles at a scan rate of 500 mV s⁻¹. Then, two CVs cycling between 0.05 and 1.00 V were recorded at 10 mV/s and used as the blank reference for Cu UPD experiments. Next, the working electrode was maintained at a certain deposition potential in an H₂SO₄ electrolyte (0.5 M) containing CuSO₄ (5 mM) for 100 s. The deposition potential was determined as the potential just above the bulk Cu deposition. Afterwards, an LSV was measured from the deposition potential to 1.00 V with a scan rate of 10 mV s⁻¹. EASA values were evaluated from the Cu desorption charge of the linear background-corrected desorption performance. The measured charge was normalized by using the theoretical value of 0.42 mC cm⁻² for a two-electron transfer assuming the oxidation of one Cu molecule to Cu²⁺ per metal atom.

To verify the accuracy of Cu UPD method, we also evaluate the surface area of **RuIr-NS** by the TEM based on a spherical geometry²⁶.

$$A_s \approx \frac{\sum \pi d^2}{\sum (1/6) \rho \pi d^3} = \frac{6 \sum d^2}{\rho \sum d^3} = \frac{6}{d_{v/a}/\rho}$$

where ρ is the bulk density of **RuIr-NS**, d is the average size determined by TEM, $d_{v/a}$ is the volume/area average diameter. We accounted for at least 200 NPs for evaluating A_s .”

Q4. The activity reported for Ru-Ir NC is rather high, higher in fact than that of the reference materials used in this work. However, the choice of such reference materials can be questionable since the actual performance of IrO_x and RuO_x nanosized particles depends on features such as particle size. In addition, the activity is similar to that reported for state-of-the-art Ru based catalysts, as shown in Table S4. As for the mass activity (or specific activity), the real figure of merit would be the activity per gram of metal (Ru+Ir) rather than the catalysts loading most of the catalysts reported in the literature are oxides, therefore containing a lower amount of Ru or Ir than the one used in this work.

Answer: Thank you for your comment. We guess that what you mentioned about is the OER catalysts listed in Table S1 rather than Table S4. Table S1 lists both bulk and nanoparticles. For example, in Ref. 26, Yang and co-authors reported RuO_x and IrO_x nanoparticles. Ref. 51 is about IrNiO_x nanoparticles. We agree with you that the actual performance of IrO_x and RuO_x depends on the features such as size. Therefore, we also showed the comparison of specific activities which was strongly related to the particle sizes in Supplementary Fig.15d. Although we cannot include all the reported catalysts, the **RuIr-NC** should belong to the catalysts with a high specific activity.

Following your suggestion, we remade Fig. 2b. The mass activities of the oxide catalysts are normalized by the weight percentage of Ru or Ir in the catalysts. Even just counting the mass of pure metal, the mass activity of **RuIr-NC** is higher as shown in Fig. 2b. In Supplementary Table 1, we also list the detailed loading amount of each catalyst for a fair comparison.

Revision 9: Page 15, Line322, we changed Fig. 2b and in the Figure caption, we added: “The mass activities are normalized by the mass of noble metals”.

Fig. 2b Mass activity of **RuIr-NC**, **RuIr-NS** and the reported high-performance catalysts with considerable stability in acid including RuO₂ NP²³, IrO₂ NP²³, IrO_x-ATO²², 3D Au-Ru²⁵, IrNiO_x⁶ and Cr_{0.6}Ru_{0.4}O₂²⁴. The mass activities are normalized by the mass of noble metals.

Q5. Cyclic voltammograms up to 1.8 V should be shown, showing several cycles. This potential has been chosen by the authors to conduct their stability study.

Answer: Thanks for your many suggestions. We have performed five continuous LSV scans up to 1.8 V (without iR correction) to conduct the stability test in Supplementary Fig. S16. We added the LSV curves in Supplementary Fig. S16.

Revision 10: In the Supplementary Information, we added LSVs up to continues 5 cycles.

Supplementary Fig. 16 Continuous OER polarization curves of catalysts and ICP analysis. **a.** Ru NPs. **b.** RuIr-NS. **c.** RuIr-NC. **d.** ICP results show the dissolved percentage of Ru or Ir in the electrolyte after the 1st, 2nd, and 5th LSV scan. Error bars were obtained by three independent ICP tests. The weight percentages were obtained by dividing the dissolved amount by the total amount on the electrode before OER. LSV scan condition: 5 mV s⁻¹, 1600 rpm. The time of the CP test for RuIr-NS and RuIr-NC is 50 min and 122 h, respectively. Please note the Ir percentages during LSV scans in RuIr-NC might not so accurate due to the low amount.

Q6. How is double layer corrected? Also, the raw voltammograms, without double layer correction should be shown. If potentials have been iR corrected, it should be indicated in the figures.

Answer: The process of how we subtract the background current and iR -correction is shown in the Supplementary Fig.12a, b. We directly subtracting the current at 1.2 V just before OER in an LSV scan.

Revision 11: Page 5, Line 130, “All the LSV curves are background current and ohmic corrected and the currents contributed from the bare RDE and carbon/Nafion mixture at higher potentials are negligible (Supplementary Fig. 12)”.

In the Supplementary Information, we added Fig. 12a, b

Supplementary Fig. 12 Ohmic and background corrections of the OER activity of **RuIr-NC**. **a**, the polarization curve of **RuIr-NC** (“Raw data”, solid red line, shown in the range of 1.15 – 1.35 V) is corrected by subtracting the background current density. The current density at 1.2 V, 0.25 mA cm⁻², is considered as the background current density which comes from the capacitance current and oxidation of metal and carbon before OER. With the background-correction, the current is almost zero before the onset of OER (dashed blue line). **b**, the ohmic-corrected OER current (dashed blue line) is then corrected with the measured ionic resistance ($\approx 38.3 \Omega$) to yield the final electrode OER activity (solid black line).

Q7. What is the resistance obtained?

Answer: The resistance is obtained at the open circuit potential by a resistance test procedure of the electrochemical station. Taking **RuIr-NC** for example, the value (ca. 38.3 Ω) is similar to the one obtained by electrochemical impedance spectra (ca. 37.5 Ω , see Figure B below). We record this resistance value and compensate the iR drop by a percentage of 85% manually (shown in Supplementary Fig. 12b). We added this information in the experimental section.

Figure B. Nyquist plot of **RuIr-NC** obtained at a potential of 1.38 V_{RHE} .

Revision 12: Page 21, Line 428, “The ohmic losses are corrected (with a percentage of 85%) based on the resistance obtained at the open circuit potential by a resistance test procedure of the electrochemical station (Supplementary Fig. 12)”

Q8. Usually, OER is studied under O_2 saturated electrodes.

Answer: Thanks for your suggestion. We have tested the OER under O_2 saturated electrolytes. The trend of activity of these catalysts is similar to under Ar-saturated electrolytes (Figure C-a). However, we found that the OER currents of the catalysts measured under O_2 -saturated electrolyte were slightly higher than those obtained under Ar-saturated electrolyte (Figure C-(b-e)). This might be because the Ru and Ir-based metallic catalysts are reactive to the O_2 in the electrolyte. The data obtained in O_2 -saturated electrolyte were added in Supplementary Fig.13a.

Figure C. **a**. Comparison of the OER activities of different catalysts. Experimental condition: O₂-saturated 0.05 M H₂SO₄, scan direction: anodic. **e-f**, comparison of the LSV obtained in O₂ (red line) or Ar (black line) saturated electrolyte.

Revision 13: In the Supplementary Information, we added Fig. 13a

Supplementary Fig. 13 OER performance in O₂-saturated 0.05 M H₂SO₄. **a**, comparison of the LSVs of the tested catalysts obtained with positive scan direction (from high potential to low potential).

Q9. What is the area of the electrodes used in the home-made electroyser?

Answer: The electrode area where the catalysts were loaded in the home-made electrolyzer is ca. $1 \times 2 \text{ cm}^2$. We added it in the experimental section.

Revision 14: Page 21, Line 431, “The catalysts were loaded on carbon paper (an area of ca. $1 \times 2 \text{ cm}^2$) with a loading amount of 0.15 mg cm^{-2} ”.

Q10. For the understanding of the durability, the authors have analysed the dissolution of Ru and Ir during repeated polarization. What is the actual program used?

Answer: Thank you for your questions. The actual program is repeating the LSV scans (1600 rpm , 5 mV/s) in OER range (from $1.1\text{-}1.8 \text{ V vs. RHE}$, without iR correction). We have added the LSV curves and detailed program in the caption of Supplementary Fig. 16.

Revision 15: In the Supplementary Information,

Supplementary Fig. 16 Continuous OER polarization curves of catalysts and ICP analysis. **a**, Ru NPs. **b**, RuIr-NS. **c**, RuIr-NC. **d**, ICP results show the dissolved percentage of Ru or

Ir in the electrolyte after the 1st, 2nd, and 5th LSV scan. Error bars were obtained by three independent ICP tests. The weight percentages were obtained by dividing the dissolved amount by the total amount on the electrode before OER. LSV scan condition: 5 mV s⁻¹, 1600 rpm. The time of the CP test for **Rulr-NS** and **Rulr-NC** is 50 min and 122 h, respectively. Please note the Ir percentages during LSV scans in **Rulr-NC** might not so accurate due to the low amount.

Q11. The experimental section states that long-term stability has been evaluated at 10 mA/cm², but in some cases the durability tests have been conducted at 1 mA/cm² (see Fig 2c), this current density is not representative for the performance of an electrolyser.

Answer: For evaluating the stability of the electrolyzer built by **Rulr-NC**, we fixed the current density at 10 mA/cm². For evaluating the stability of OER catalysts, we use a CP test under 1 mA/cm². This is mainly because **Rulr-NS** quickly lost its activity even at an OER current density of 1 mA/cm². However, based on the suggestion from both you and Referee 1, we also tested the OER stability of **Rulr-NC** at 10 mA/cm². The **Rulr-NC** is stable for at least 40 h, which is better than the other catalysts list in Table 1.

Revision 16: Main text, Page 7, Line 164: “The **Rulr-NC** can sustain 40 h even under 10 mA cm⁻²_{geo} (Supplementary Fig. 17)”.

In the Supplementary Information, we added Fig. 17 and modified in Table 1.

Supplementary Fig. 17. CP curve of **RuIr-NC** at the OER current density of $10 \text{ mA cm}^{-2}_{\text{geo}}$. The **RuIr-NC** can sustain its activity for more than 40 h (with less than 5% increase of the potential).

Q12. After durability tests, it is concluded that Ru-Ir NC has very high stability because had only 25% and 15% of Ru and Ir loss, respectively, even after five scans. Actually, this is a severe loss of material for only 5 cycles. In addition, this test fails short to assess material's durability since in many cases, especially with Ru based catalysts, activity drop faster with the increasing number of scans.

Answer: The repeating cycling is a more harsh experimental conditions for evaluating the stability of catalysts rather than the CP test. We agree with you that 25% and 15% of Ru and Ir loss is still severe as a practical catalyst. However, compared to the result of **RuIr-NS**, we believe that this improvement in the anti-dissolution ability is a great step for Ru-based catalysts. From the ICP measurement, we wanted to emphasize that the anti-dissolution ability of **RuIr-NC** is much higher than that of **RuIr-NS**. Besides, the dissolved amount of Ru and Ir in **RuIr-NC** and its activity did not change so much after the 2nd cycles, while **RuIr-NS** kept dissolving and the losses reached about 90%. We keep on studying more stable OER catalysts based on these findings of **RuIr-NC**.

Q13. The bulk composition of both Ru-Ir particles seems to be similar, but what about the surface composition, which is actually more relevant for the catalytic performance. What is the Ru/Ir surface atomic ratios in both samples? What are the oxidation states of surface Ru and Ir in both samples?

Answer: We use the lab XPS to reveal the Ru/Ir surface atomic ratios of all the samples. Considering the source energy of Mg K α , the lab XPS has a probe depth of approximately 1 nm. Therefore, the surface composition of NPs can be investigated by integrating their corresponding XPS peaks. The surface molar ratio of Ru: Ir for **RuIr-NC** and **RuIr-NP** is 0.935: 0.065 and 0.920: 0.080, respectively. The binding energy of each orbital in **RuIr-NC** and **RuIr-NS** is quite similar to the HAXPES results (Supplementary Fig.19 and Table A

below). This result suggests that the surface Ru and Ir atoms in both samples mainly in a metallic state with the same composition.

Table A Binding energies of Ru and Ir from lab XPS and HAXPES

Orbital	Lab XPS ^[1]		HAXPES of Rulr-NC/Rulr-NS
	Rulr-NS	Rulr-NC	
Ru3P _{3/2}	461.8	461.7	461.7
Ru3P _{1/2}	484.0	483.9	483.8
Ir4f _{7/2}	61.1	60.9	60.8
Ir4f _{5/2}	64.1	63.9	63.9

[1] the resolution is 0.1 eV.

Revision 17: Page 8, Line 195, “Lab XPS results show that the surface composition and electronic structure of Ru and Ir in the two Rulr catalysts are similar as well (Supplementary Fig. 19 and Table 4)”.

Supplementary Fig.19: Comparison core-level spectra between lab XPS and HAXPES results. **a**, Ru3p spectra. **b**, Ir4f spectra.

Supplementary Table 4 The atomic percentage (%) of Ru and Ir in the as-prepared Rulr catalysts and Rulr catalysts during OER.

	As-prepared	EDX @ JEOL ARM-200 kV			XRF As-prepared	XPS As-prepared
		1.2 V	1.4 V	1.8 V		
Rulr-NS	95.9 / 4.1	94.5 / 5.4	95.2 / 4.8	83.8 / 16.2	93.5 / 6.5	92.0 / 8.0
Rulr-NC	96.0 / 4.0	96.0 / 4.0	93.8 / 6.2	92.8 / 7.8	93.6 / 6.4	93.5 / 6.5

Q14. XAS analysis identifies RuO₂ in the sample subjected to 1.8 V. Have the authors observed Ru (and Ir) atoms in upper oxidation states? Ru tends to form perruthenates at such high potentials. The formation of oxides, and their actual nature, is very relevant for the activity of Ru and Ir based catalysts for the OER. Is indeed the development of an amorphous oxide layer observed by STEM related to the OER activity of the samples?

Answer: Thanks for your question. We did not observe crystalline RuO₂ by XRD and HAADF-STEM. The metal (hydro)oxides generated during OER is amorphous. In Fig.3a, we just use the XAS spectra of standard RuO₂ for LCF analysis because only RuO₂ and Ru are acceptable as the components of LCF (Supplementary Fig. 27).

We also expected that upper oxidation states of Ru would be generated during the reaction. Therefore, we tested the XAS of RuO₄. However, RuO₄ is not an acceptable component of LCF, indicating that there are undetectable RuO₄ in our case as shown in Supplementary Fig. 27d.

Q15. Subjecting Ir and Ru to high potentials in acid electrolyte would most like lead to material's corrosion, which result in strong positive currents. What fraction of the current reported in this work accounts to the OER and what fraction accounts to materials corrosion?

Answer: Thanks for your question. To figure out the fraction of OER, we tested the Faradic efficiency of **RuIr-NC**. The Faradaic efficiency of the **RuIr-NC** was 98.5% at 20 mA cm⁻², which suggests that the oxidation current is mainly derived from OER. We added the figures and experimental details in the Supplementary Information.

Revision 18: Main text, Page 6, Line 141, "The Faradaic efficiency of the **RuIr-NC** was 98.5% at 20 mA cm⁻² which suggests that the oxidation current is mainly derived from OER (Supplementary Fig. 15)".

Page 21, Line 435, "For the Faradaic efficiency test of **RuIr-NC**, we monitored the generated oxygen amount by gas chromatography (GC) after fixing the electrode at each

current density for 30 min. An H-cell and Nafion 117 film were used to separate the anodic and cathodic chamber.”

In the Supplementary Information, we added Fig. 14.

Supplementary Fig. 14 **Faradaic efficiency of RuIr-NC**. **a**, calculated O₂ and measured O₂ amount by GC. **b**, Faradaic efficiency of **RuIr-NC**. The current density is not background corrected, and therefore the lower Faradaic efficiency at small current densities is caused by the oxidation of metal and carbon/PVP/Nafion.

Q16. The conclusion that a cheaper electrolyser has been constructed is not sustained. What is the price of each electrolyser? Moreover, catalysts are not the most expensive components of PEM water electrolysers.

Answer: Thanks for your suggestion. We roughly calculated the prices of catalysts used in the electrolyzer based on the mass of Ru, Ir and Pt. The prices of these metals in the market were shown in Table B. We use 0.3 mg metals for an either anodic or cathodic reaction. Hence, the price of the catalysts in **RuIr-NC || RuIr-NC** electrolyzer is 0.0081 USD. While the price of the catalysts used in Pt || IrO₂ electrolyzer is 0.025 USD, which is 3 times higher than that of **RuIr-NC || RuIr-NC** electrolyzer. Following your comments, we modified the introduction.

Table B. Average prices of metals (USD/gram) for the last 1 year (Aug 2019-Aug 2020). (Information from InfoMine MaIT. <http://www.infomine.com/>)

Metal	Price (USD/gram)
Pt	30.45
Ir	52.89
Ru	8.68

Revision 19: Page 1, Line 21, “Splitting water electrochemically is among the most commonly used techniques; however, it accounts for only 4% of global hydrogen production. One of the reasons is the high cost and low performance of catalysts promoting the oxygen evolution reaction (OER)”.

Reviewer #3 (Remarks to the Author):

In this manuscript, the authors report the synthesis of RuIr nano-coral structures and their electrocatalytic performance for OER, HER and overall water splitting. They have carried out very comprehensive structural, compositional and electrochemical characterization, and demonstrated excellent performance of the RuIr-NC with respect to other control catalysts. However, there are many incorrect statements and some improper practices which make the data presented in this manuscript not fully convincing. For these reasons, the reviewer does not recommend the Editor to accept the present manuscript for publication in Nat. Commun. The reviewer's specific comments are summarized below:

Q1. There are some statements that are incorrect. For instance:

(1) Abstract: "predominately because of the high cost and low performance of catalysts promoting the oxygen evolution reaction". This is not the major reason that can explain the small share of electrolysis-derived hydrogen in the market. In fact, the cost of catalysts only accounts for a small portion of the system costs of electrolyzers (see e.g. <https://www.nrel.gov/docs/fy19osti/72740.pdf>).

(2) Line 48, page 2: the authors claim that "the design principles for their use in acid have not been well developed". In fact, there are already several review articles available about OER catalysts in acids (e.g. Adv. Energy Mater. 2016, 1601275), where how to design and optimize catalysts to achieve high OER performance is pointed out.

(3) Lines 136-137: This statement is not correct. The mass activity, by definition, is derived simply by normalizing the catalytic current to the catalyst mass. It has nothing to do with EASA. The specific activity is closely linked to EASA.

(4) Lines 241-243: Bifunctional catalysts have been investigated extensively in recent years. Many noble metals based OER catalysts also show reasonably good HER performance.

Answer: Thank you very much for your carefully checking and sharing us with many nice papers. In particular (3), the sentence we previously wrote was misleading. We calculated

the mass activity by using the mass of metals in the manuscript. We corrected these places and added some references as follows.

Revision 20: (1) Page 1, Line 21, we modified “predominately because of” to “One of the reasons is”

(2) Page 2, Line 47, we cited this references in “there are only a few reports referred to the design principles for the catalysts used in acid¹²”

(3) Page 6, Line 148, we modified the sentence as “The specific activity is known to be fundamentally determined by the catalysts’ EASA”

(4) Page 11, Line 263, “Bifunctional catalysts for both HER and OER are highly desirable for practical use as an integrated water electrolyzer. Besides, Ru and Ir metals are also reported to catalyze HER³¹. The HER properties of **RuIr-NC** were investigated by using RDE in the same electrolyte as OER”

Q2. Line 37, page 2: what does “are ready at ...” mean? Practical PEM water electrolysis operates at a current density of 1 A/cm² or above, are the authors sure that only millivolts are needed in this case?

Answer: Thank you for pointing out this. Our sentence was misleading. We wanted to emphasize that the potential loss of HER compared to that of OER is tiny. We changed the sentence as follows.

Revision 21: Page 2, Line 36, “HER is kinetically favoured in acid because of the high proton concentration, where high current densities such as 10 mA/cm²_{geo} are ready even at (ten of) millivolts of overpotential with a negligible amount of Pt catalysts³”.

Q3. Line 65, page 3: “practical current density” appears several times in the manuscript. Can the authors specify what they think is the practical current density? For PEM water electrolysis, a practical current density should be 1 A/cm² or even higher. For solar water splitting, the current density is usually in the range of 10 – 20 mA/cm².

Answer: Thanks for your suggestions. We modified the places when we use “practical current density”. We listed them as following.

Revision 22: Page 2, Line 36, “HER is kinetically favoured in acid because of the high proton concentration, where a high current density (such as $10 \text{ mA/cm}^2_{\text{geo}}$) is ready even at millivolts of overpotential with a negligible amount of Pt catalysts”

Page 3, Line 65, “The obtained **Rulr-NC** is also matched with remarkable stability that it showed no noticeable degradation throughout 122 h under a fixed current density of 1 mA cm^{-2} ”

Q4. The authors mention several times “the known systems”. This is confusing. According to the work reported here, they are not working at a “system”(electrolyzer) level.

Answer: We changed “the systems” to “overall water splitting cell”

Revision 23: Page 1, Line 26, and Page 3, Line 68, “A home-made overall water splitting cell using **Rulr-NC** as both electrodes can reach $10 \text{ mA cm}^{-2}_{\text{geo}}$ at 1.485 V for 120 h without noticeable degradation, which outperforms known cells”

Page 3, Line 71, “One should note that the as-built overall splitting cell is significantly less expensive than the combination of commercial Pt and IrO_x catalysts.”

Page 21, Line 430, “The overall water splitting experiments were conducted in a homemade two-electrode overall water splitting cell with **Rulr-NC** as both the anode and cathode.”

Page 17, Line 337, the caption for Figure 4, “overall water splitting cells”.

Q5. Page 4: discussion about XRD data – parentheses should be used when mentioning the indexed facets (lines 96-97)

Answer: Thank you for pointing out this. We added parentheses when indexing the facets.

Revision 24: Page 4, Line 97, “The crystal size estimated from the sharp peaks of (10 $\bar{1}$ 0) and (11 $\bar{2}$ 0) were 15.2(3) and 13.4(5) nm, respectively, while that from (0002) was 3.1(2) nm”

Q6. The authors claim that there are 4 at% Ir in the RuIr-NC according to the EDS result. What spectral lines (L or M) did they use for the quantitative analysis? An EDS spectrum should be given. Moreover, according to the signal intensity of Figure 1h, it does not look that the atomic percentage of Ir is only 4%. Did the authors manipulate the intensity manually in the software?

Answer: We use Ru-L and Ir-M for quantitative analysis. We added the information in Figure 1 caption. An EDS spectrum was added in the Supplementary Fig.10. All maps in Fig.1 are original experimental data. We used NORAN System 7 software developed by Thermo Fisher scientific for EDS analysis. The colour tone of images was automatically assigned by the software. Also, the elemental maps are count maps for each element rather than composition maps. Each map was independently scaled to the highest intensity of each image. If the maps were atomic composition maps, Ir mapping image should be darker than Ru mapping image and is difficult to be seen. This is not the case in our EDS analysis.

Revision 25: Page 13, Line 315, the caption of Fig.1, “showing **g**, Ru-L, and **h**, Ir-M”.

In the Supplementary Information, we added Fig. 10.

Supplementary Fig.10. EDS spectrum of the particle shown in Fig.1b. Clear Ru and Ir peaks were shown. The Fe and Co, and Si signal came from a pole piece of TEM magnetic electron lens and STEM grid, respectively.

Q7. The authors claim that (0001) crystal face is exposed. However, from Figure 1c, there are obviously at least two different regions where the atoms are arranged in different ways. How do the authors explain this?

Answer: **Rulr-NC** is not a single crystal but an assembly of nanosheets which grow to different directions as shown in Fig.1d. In this case, only when (0001) facet is perfectly perpendicular to the electron beam, we can observe the clear hexagonal pattern. If a sheet is viewed from different directions or overlapped with another sheet, 2D STEM image shows different atomic arrangements or blurred patterns. In the case of Fig.1c, the blurred part at the bottom of the image might result from an overlapped another sheet. However, XRD, 3D-tomography and HAADF-STEM images showed that (0001) is mainly exposed in **Rulr-NC**.

Q8. Figure 2a: the data showing here are questionable. Where couldn't Ru NPs reach 10 mA/cm²? It seems that the authors swept the potential positively, which is not a proper practice for OER assessment because in this case the oxidation from redox will appear leading possibly to an overestimation of the activity. The potential should be swept negatively in order to avoid the overestimation. Furthermore, the LSV of Ru NPs is deformed, which may result from the over iR-correction. The degree of iR-correct is not given in the manuscript.

Answer: According to your suggestion, we have tested the negative scan LSV (shown in the figure below). We found the overestimation of activity of Ru NPs in the positive scan as you mentioned. However, for **Rulr-NC**, there is little difference between positive and negative LSVs, which also supports our conclusion that **Rulr-NC** has high stability. Still, most reports in the literature used positively scanned LSVs. To compare with them, we

use positive scan data in the main text. We added the negative scan data in the Supplementary Fig. 13 and mentioned it in the main text.

The degree of iR -correction is 85% we added this in the Supplementary Information. For Ru NPs, the raw LSV *data* also did not show 10 mA cm⁻² during the test area (1.2 – 1.8 V, Figure D below). The LSV of Ru obtained from the positive scan shows a peak. This is because of the metal dissolution. This LSV shape of the Ru NPs is similar to the other reported Ru nanocatalysts (one example is presented in the solid black line, Figure E below).

Figure D The raw LSV of Ru NPs.

Figure E: First quasi-stationary OER scan for bulk (dashed line) and nanoparticle catalysts (solid line) of Ru, Ir, and Pt recorded with 6 mV/s and 1600 rotations per minute (rpm) in deaerated 0.1 M HClO₄ at RT. The current is normalized to the number of surface sites determined from CO stripping experiments. (Ref.19, Tobias Reier, *ACS Catal.* 2012, 2, 8, 1765–1772)

Revision 26: Page 6, Line 139, we added “The high OER activity of **RuIr-NC** can be also reflected by the almost overlapped anodic and cathodic LSVs (Supplementary Fig. 13).”

In Supplementary information, we added Fig. 13.

Supplementary Fig. 13 OER performance in O_2 -saturated 0.05 M H_2SO_4 . a, comparison of the LSVs of the tested catalysts obtained with positive scan direction (from high potential to low potential). OER performance tested with negative scan direction (from high potential to low potential). a, comparison of the LSVs of the tested catalysts. b-e, comparison of the LSVs obtained by positive (blue line) and cathodic (black line) scan direction.

Q9. The Rulr-NS and Rulr-NC were synthesized using the same procedure. Then under which conditions can one obtain Rulr-NC and which other conditions to obtain Rulr-NS? Why? This should be discussed in more detail in the manuscript.

Answer: The procedures of synthesizing **Rulr-NC** and **Rulr-NS** are slightly different from each other. **Rulr-NC** was obtained by a hot-injection method, that is adding the metal precursors solution to the preheated TEG solution at 230 °C. **Rulr-NS** was obtained by a heat-up method, that is heating the TEG solution containing the metal precursors from RT to 230 °C. In a hot-injection process, it is known that the rate of surface atom diffusion is very high which prompts the exposure of facet with lower surface energy. (Ref. 24, Xia, X. et al., *Proc. Natl. Acad. Sci.* **110**, 6669-6673 (2013)) Therefore, the hot-injection method would realize **Rulr-NC** mainly exposing the lowest energy (0001) facets. (Ref. 25, Liu, J. et al., *J. Am. Chem. Soc.* **135**, 16284-16287 (2013))

Revision 27: Page 5, Line 119, we added “**Rulr-NS** was synthesized by heat-up method, that is heating the mixture of metal precursors in TEG and PVP solution from RT to 230 °C. In contrast, **Rulr-NC** is obtained through a hot-injection process. It is known that the rate of surface atom diffusion is very high at an elevated temperature, which prompts the exposure of facet with lower surface energy.²⁴ Therefore, the hot-injection method would realize **Rulr-NC** mainly exposing the lowest energy (0001) facets.²⁵”

Q10. The authors failed to specific the loading mass of each catalyst during the HER or OER test, which influences the geometric activity.

Answer: The loading amount of our catalysts were noted in the Supplementary Table S1. To clarify more clearly, we added detailed information in the Experimental Section. The loading amount of the catalysts at RDE electrode is 0.05 mg_{metal}/cm².

Revision 28: Page 20, Line 414, we added “The catalyst inks were prepared by loading the as-prepared nanoparticles onto carbon (Vulcan XC-72R) with a metal weight percentage of approximating 20 % (determined by elemental analysis). Then, the carbon-loaded catalysts (5 mg in total mass) were dispersed in 1 ml of a mixture solution of isopropanol (0.600 ml), water (0.300 ml), and 5 wt% Nafion (0.100 ml). Then, the catalyst ink (10 µl) was dropped onto the surface of the working electrodes (0.05 mg_{metal}/cm²) and dried under air overnight. For comparison, commercial Pt/C catalyst inks were prepared by following the same process.”

Q11. Lines 140-142: The reviewer does not agree with this statement. For Cu UPD, it is likely that Cu cannot be deposited on every catalytically active sites because of the mass transfer limitation. In this case, the real catalytically active surface is in fact underestimated, while the specific activity is overestimated. The thus-obtained specific activity is the maximal possible one. It seems that the authors mix up the mass activity and specific activity. The fact that “the total mass content of metals” is considered actually verifies that what the authors state is wrong.

Answer: Thank you for your questions. The evaluation of the real surface area of the catalyst is still an open issue. To get a reliable ECSA, we seek for several methods. We have tried CO stripping away method. However, this method is not suitable because the conversion factor, 0.42 mC cm^{-2} , is not adaptable for Ru metal. Also, Referee 2 pointed out the disadvantage of this method. Several papers suggest that Cu UPD is more adaptable for most platinum-group metals compared to other methods such as CO oxidation or BET area. Therefore, we evaluate the EASA values by using Cu UPD method, in which a conversion value of 0.42 mC cm^{-2} has been used for both Ru and Ir (Ref. 44, *Inter. J. Electrochem. Sci.* 11, 4442-4469 (2016).) (Supplementary Fig. 12a and b). Moreover, to confirm whether the Cu UPD method is reasonable, we also evaluated the EASA value of **RuIr-NS** based on the size analysis from TEM based on a spherical model (*J. Electrochem. Soc.* 152, A2256-A2271 (2005) and *J. Phys. Chem. Lett.* 3, 399-404 (2012)). The EASA values of **RuIr-NS** obtained by two methods are quite similar (114 vs. 111 m^2/g , Supplementary Fig. 12c). Also, about the sentences describing “total mass content of metals”, similarly as what we answered for Question 1(3), our description in the sentence is very misleading. We delete this sentence.

Revision 29: Page 6, we delete “Although not all the catalysts on RDE takes part into OER, we estimated the surface area of each catalyst by using the total mass content of metals and obtained maximum active sites and minimum specific activity, which can be considered as a fair approximation.”

Q12. The authors assessed the stability at a low current density of 1 mA/cm^2 in such a way they've observed a long catalytic stability of 122 h. Why didn't they perform the test at 10 mA/cm^2 ? For PEC water splitting, 10 mA/cm^2 is a sensible current density; while for electrocatalytic water splitting, catalysts should be able to sustain at much higher current densities for a long term.

Answer: Thank you very much for your suggestion. We chose the current density of 1 mA cm^{-2} because the **RuIr-NS** lost its activity too fast (within minutes) under 10 mA cm^{-2} . Following your suggestion, we performed the OER stability measurement of **RuIr-NC** at 10

mA cm^{-2} and added it in the Supplementary Fig.17 and Table 1. The **Rulr-NC** is stable for at least 40 h, which is better than the other catalysts list in Table 1.

Revision 30: Main text, Page 7, Line 164: “The **Rulr-NC** can sustain 40 h even under 10 $\text{mA cm}^{-2}_{\text{geo}}$ (Supplementary Fig. 17)”.

In the Supplementary Information, we added Fig. 14 and modified in Table 1.

Supplementary Fig. 14. CP curve of **Rulr-NC** under the OER current density of 10 $\text{mA cm}^{-2}_{\text{geo}}$. The **Rulr-NC** can sustain more than 40 h (with less than 5% increase of the potential).

Q13. *In Table S1, the authors overlooked some high-performance Ir based OER catalysts developed recently. They are recommended to carefully check recent publications and included the latest advances for comparison.*

Answer: Thanks for your suggestion. We carefully check the latest high-performance Ir based OER catalysts and added them to the references. We also surveyed Ru-based catalysts and added in Table 1.

Revision 31: In Supplementary Table 1, we added References 71-74

Q14. *Lines 170-176: The Rulr-NC lost its Ir and Ru by 15% and 25%, respectively, only after a few polarization scans, which are significant actually. This implies that*

at a high current density (potential) Rulr-NC is likely unstable. That was probably why the stability was only assessed at 1 mA/cm².

Answer: We agree with you that 25% and 15% of Ru and Ir loss is still severe as a practical catalyst. However, compared to the result of **Rulr-NS**, we believe that this improvement in the anti-dissolution ability is a great step for Ru-based catalysts. From the ICP measurement, we wanted to emphasize that the anti-dissolution ability of **Rulr-NC** is much higher than that of **Rulr-NS**. Also, the dissolved amount of Ru and Ir in **Rulr-NC** and its activity did not change so much after the 2nd cycles, while **Rulr-NS** kept dissolving and the losses reached to about 90%. We keep on studying more stable OER catalysts based on these findings of **Rulr-NC**

We also do agree with you that **Rulr-NC** becomes less stable under high potential. This is the nature of all-metal catalysts. We showed 1 mA cm⁻² mainly because the **Rulr-NS** loses its activity within several minutes. However, based on the suggestion from you and the other two referees, we performed the stability test of **Rulr-NC** by CP at 10 mA cm⁻². **Rulr-NC** is stable for at least 40 h, which is better than the other catalysts listed in Table 1. We added this data in the Supplementary Fig.17 and Table 1. The data can be seen in the answer for Q13.

Q15. Lines 190-192: The authors should briefly mention how the isotropic Ru-Ir catalyst was prepared in the main text.

Answer: We added a sentence to describe the preparation of **Rulr-NS** and **Rulr-L**.

Revision 32: Page 5, Line 119: **Rulr-NS** was synthesized by heat-up method, that is heating the mixture of metal precursors in TEG and PVP solution from RT to 230 °C.

Page 8, Line 202: An isotropic Ru-Ir catalyst was prepared by a similar heat-up method to that of **Rulr-NS** and subsequent high-temperature annealing under vacuum.

Q16. Fig. S18: From the TEM micrograph, the size of Rulr-L is significantly increased. This likely explains why Rulr-L has poorer performance. This is

**contradict with the authors' claim that "the crystal size is not a determined reason".
What is the EASA of RuIr-L?**

Answer: Following your suggestion, we measured EASA of **RuIr-L** and attached the results in the Supplementary Fig.23. Although **RuIr-L** has a large particle size, the crystal size of **RuIr-L** is only 14.2 nm which is similar to that the **RuIr-NC**. Also, it is known that the surface areas of metal particles are not largely different if the particles sizes are above 10 nm because the surface-to-volume ratio of 10 nm nanoparticles is approximately less than 10%. Therefore, **RuIr-L** has quite a similar crystal size (14.2 nm) and ECSA (31.3 m²/g) to **RuIr-NC**, and **RuIr-NC** shows better performance than **RuIr-L** in terms of both geometric activity and specific activity.

Revision 33: In the Supplementary information, we modified Fig.23

Fig. 23 OER performance of RuIr-L. a, the geometric activity of **RuIr-L. b,** Chronopotentiometric curves of **RuIr-L** at a current density of 1 mA/cm⁻². **c,** EASA evaluation of RuIr-L based on Cu UPD. **d,** the specific activity of **RuIr-L. RuIr-NC** and **RuIr-NS** were used for comparison.

Although **Rulr-L** has a similar crystal size to that of **Rulr-NC**, **Rulr-L** requires a much higher overpotential to achieve $10 \text{ mA cm}^{-2}_{\text{geo}}$. Also, the CP result in Fig. S23b suggests much lower stability than that of **Rulr-NC**. Considering the specific activity, **Rulr-NC** is higher than both **Rulr-L** and **Rulr-NS**.

Q17. *In Figure 4d inset and the movies S1&S2, the authors emerged the electrode holders into the acidic solution for testing, which is perhaps not proper practice. What is the material of electrical contact? If it is Pt, it may contribute to the electrocatalytic process as the solution can easily access to the Pt sheet. Corrosion might also happen in this case. The electrical contact should be protected from the attack of corrosive acidic electrolyte (using either epoxy resin or simply lifting the holder to a place far away from the electrolyte surface).*

Answer: Thanks for your nice suggestions. The electrical contact is Pt. Following your suggestion, we retested the overall water splitting experiments. We revised Figure 4d and attached a new movie.

Revision 34: Page 24, Fig.4d is replaced

In the Supplementary Information, Movie 2 is replaced.

Fig. 4 d, Chronopotentiometric curves of **Rulr-NC || Rulr-NC** at a current density of $10 \text{ mA cm}^{-2}_{\text{geo}}$. The inset shows the two-electrode configuration with bubbles on both electrodes.

Q18. Fig. S8: Histograms showing the size distribution of the samples should be given as insets.

Answer: The histograms showing the size distribution were added.

Revision 35: In the Supplementary Information, we added the histograms to Fig. S8.

Fig. 8 Large area, bright-field TEM images of synthesized nanoparticles. a, Rulr-NS. b, Ru NPs, and c, Ir NPs. d-f, the corresponding histograms showing the size distributions of a-c), respectively. The scale bars are 50 nm.

Q19. How was the catalyst ink prepared? From Fig. S27, it seems that the Rulr-NC was loaded on carbon support?

Answer: Yes, the as-prepared NCs or NPs were loaded on carbon. We added ink preparation in the experimental section.

Revision 36: Page 20, Line 414, “The catalyst inks were prepared by loading the as-prepared nanoparticles onto carbon (Vulcan XC-72R) with a metal weight percentage of approximating 20 wt% (determined by elemental analysis). Then, the carbon-loaded

catalysts (5 mg in total mass) were dispersed in 1 ml of a mixture solution of isopropanol (0.600 ml), water (0.300 ml), and 5 wt% Nafion (0.100 ml). Then, the catalyst ink (10 μ l) was dropped onto the surface of the working electrodes (0.05 mg_{metal}/cm²) and dried under air overnight. For comparison, commercial Pt/C catalyst inks were prepared by following the same process.”

Q20. Lines 248-249: Given the Tafel slope, both Ru and Ir NPs should follow a Volmer-Heyrovsky mechanism during the HER. The Volmer mechanism works when the Tafel slope exceeds 120 mV/dec. BTW, it should be “Heyrovsky”, rather than “Heyrosky”.

Answer: Thanks for your carefully check. We corrected these sentences.

Revision 37: Page 11, Line 269, “Ru or Ir NPs shows a Tafel slope of 81.0 mV/dec or 42.8 mV/dec (Fig. 4b), respectively, which suggests a Volmer-Heyrovsky mechanism (Volmer: $H^+ + e^- \rightarrow H_{ad}$, 120 mV/dec; Heyrovsky step: $H_{ad} + H_2O + e^- \rightarrow H_2 + OH^-$, 40 mV/dec in the theoretical calculation).”

Q21. Line 259: I would suggest the authors use “practical” with caution. With these “practical” catalysts, PEM water electrolysis can be accomplished at 500 mA/cm² under ca. 1.5 V (see e.g. J. Electrochem. Soc. 2018, 165, F305), much better than the Rulr-NC bifunctional catalysts.

Answer: Thank you for your suggestions. We revised the several sentences in the manuscript containing ‘practical’. We listed them below.

Revision 38: Page 3, line 65, “no noticeable degradation throughout 122 h to deliver a practical current density” was changed to “no noticeable degradation throughout 122 h under a fixed current density of 1 mA cm⁻²”.

Page 11, Line 286, “the practical catalyst combination of IrO₂ || Pt/C” was changed to “the combination of the commercial catalysts IrO₂ || Pt/C”

Q22. The authors are recommended to improve their English. There are a number of grammatical errors, typos and inappropriate usage of English. Just to name a few:

Line 53, page 3: what does “heave Ir doping” mean? Should it be “heavy”?

Line 69, page 3: “operating” should be inserted after “keeps”

Lines 95-96, page 4: instead of “...size...were...”, it should be “...sizes...are...”

Line 103: what do the squares mean?

Line 139: it should be “...catalysts... take...”, rather than “takes”

Line 142: According to Fig. S12d, the specific activity should be 4.9 mA/cm², but not 4.9 A/cm²

Line 170: “valid” should be corrected to “validate”

And many others...

Answer: Thank you very much for your carefully check. We corrected these places and carefully checked other places in the manuscript.

Revision 39: Line 53, page 3, “heave” was corrected to “heavy”

Line 69, page 3: “operating” was corrected to “keep operating”

Lines 95-96, page 4: “...size...were” was corrected to “...sizes...are...”

Line 103: “squares” was corrected to “rectangle”

Line 139: “takes” was corrected to “take”

Line 142: “4.9 A/cm²” was corrected to “4.9 mA/cm²”

Line 170: “valid” was corrected to “validate”

Line 62: “extremently” was corrected to “extremely”

Line 78: “metal” was corrected to “the metal”

Line 138: “acheving” was corrected to “achieving”

Line 207: “determined” was corrected to “determining”

Line 298: “aslo” was corrected to “also”

Line 303: “daemonstrates” was corrected to “demonstrates”

Line 371: “were” was corrected to “was”

Line 412: “a RDE” was corrected to “an RDE”

Line 423: “hundreds” was corrected to “hundred”

Suggestion from the editor:

In addition, we noticed that a platinum counter electrode was used in this work. Given that there is a substantial amount of concern regarding the use of such counter electrodes, especially for assessing H₂ evolution (for example, <https://www.sciencedirect.com/science/article/pii/S0378775319316283?via=ihub>). As such, we discourage the use of platinum electrodes and suggest data be collected with an inert electrode (or, alternatively, that the manuscript demonstrates no Pt contamination to occur).

Answer: Thanks for your suggestion. We have changed the reference electrode to the carbon rod when we do HER. The HER activities of the catalysts did not change with alternating the reference electrode (as shown in Figure F below). In the experimental section, we mentioned the reference electrode. Figure 4a was also revised by the curved obtained using glassy carbon rod.

Figure F: HER curves of the catalysts using glassy carbon rod (red solid line) and Pt wire (black dash line) as a reference electrode.

Revision 40: Page 20, Line 411, “The OER and HER were performed with an RDE (ϕ 5.00 mm, 0.196 cm²) coupled with Ag/AgCl reference electrode and Pt wire (for OER) or graphite rod (for HER) counter electrode using a potentiostat (CHI 760e, USA).”

Fig. 4 HER and overall water splitting performance. a, HER polarization curves. The values are the overpotential to reach $10 \text{ mA cm}^{-2}_{\text{geo}}$.

REVIEWER COMMENTS

Reviewer #1 (Remarks to the Author):

The authors have done many improvements of the manuscript and covered several of the raised questions. I remain concerned with the Comment 4 of Reviewer 1. Supplementary Fig.16 (Now Fig. 20) showed oxidation states of Ir is less than 4+, which is contradictory with the published paper Nature Catalysis 1, 841-851 (2018). The authors responded that they believe that the active species with higher oxidation states will be obtained just before or during OER. The authors should show evidence from their in situ data. Currently, Fig.20 shows low oxidation states, rather than higher oxidation states.

Reviewer #2 (Remarks to the Author):

The authors have addressed all my comments properly and the text has been modified accordingly. The paper can be published as is.

Reviewer #3 (Remarks to the Author):

The authors have made substantial effort to improving the manuscript and addressed the concerns raised by reviewers. However, there are still some improvements that should be made before the manuscript reaches the publication criteria of Nat. Commun.

1. There are still many grammatical errors to be corrected, even in the revisions made by the authors.

2. Response to the reviewer's second comment (Q2):

The revision made in Line 26, page 2 is not proper. 10 mA/cm²geo cannot be deemed as a high current density at all for an electrolyzer. The authors are recommended to revise the sentence as "..., where current densities such as 10 mA/cm²geo can be readily achieved even at ..."

Note that "ready" is not the adjective form of "readily". The authors should make clear the meanings of these words.

3. Response to the reviewer's eighth comment (Q8):

The reviewer does not agree with the authors' logic. There are many improper practices and therefore unconvincing data in the literature. This should not become an excuse for all others to follow the incorrect practices in their research work. Two wrongs do not make a right. The authors should try their best to implement proper practices in their work, particularly for the work to be published in high-IF journals. Otherwise, more researchers will be misled in future. This misleading information will jeopardize the whole research community.

4. Response to the reviewer's 12th & 13th comment (Q12 & Q13):

In Table S1, the authors only list the Ru or Ir based OER catalysts exhibiting performance poorer than that of their own RuIr-NC catalysts. This is not a proper practice. They should also make comparison with those having better performance than that of RuIr-NC, for example those reported in ACS Appl. Energy Mater. 2020, 3, 3746, where the nanoporous IrO₂ can sustain at 10 mA/cm² for 40 h as well and even at 100 mA/cm² for 30 h.

5. Response to the reviewer's 16th comment (Q16):

The reviewer does not doubt that RuIr-L shows poorer performance compared to RuIr-NC. The reviewer's point is that the crystal size matters here and therefore the authors should revise their statement that "the crystal size is not a determined reason". The ECSA of RuIr-L is only 31.3 m²/g, nearly four times less than that of RuIr-NC (114 m²/g), how can the authors claim that they have "quite a similar ECSA"??

All statements must be made and supported by solid experimental data and theoretical analysis.

Point-to-point response sheet

Reviewer #1 :

The authors have done many improvements of the manuscript and covered several of the raised questions. I remain concerned with the Comment 4 of Reviewer 1. Supplementary Fig.16 (Now Fig. 20) showed oxidation states of Ir is less than 4+, which is contradictory with the published paper Nature Catalysis 1, 841-851 (2018). The authors responded that they believe that the active species with higher oxidation states will be obtained just before or during OER. The authors should show evidence from their in situ data. Currently, Fig.20 shows low oxidation states, rather than higher oxidation states.

Answer: Thank you very much for your further explanation. We now understand your meaning. Our previous answer might be misleading. Supplementary Fig.20 shows the XAFS of Ir L_3 -edge of the as-prepared samples which have not been subjected to any electrochemical process. Therefore, the oxidation state of Ir is mainly in a metallic state, which can be also supported by the XPS data shown in the Supplementary Fig. 18b and 19b.

As we shown in the STEM analyses (Figure 3 in the main text), the **RuIr-NC** is composed of metallic core and amorphous oxide thin layer on the surface after OER (1.8 V), while **RuIr-NS** is almost fully oxidized. Therefore, we infer that Ir and Ru in the amorphous layers might have higher oxidation states and are active for OER. However, we have to note that there are only 6 at.% of Ir in our RuIr catalysts. The main catalytic species could be Ru because Ru is a more OER active than Ir (Also discussed in Ref.19, Tobias Reier et al., *ACS Catal.* 2, 1765–1772, 2012) and the Ru/Ir ratio in **RuIr-NC** did not change so much after OER.

However, following your suggestion, we performed in-situ XANES on Ir L_3 -edge of RuIr catalysts. It is known that the metallic Ir changes to Ir^{4+} in the range of 0.6-1.0 V, a potential lower than the OER thermal-equilibrium potential (1.23 V) [A. Minguzzi et al., *Chem. Sci.*, 5, 3591-3597, 2014]. Therefore, we infer that Ir^{4+} species already exist in the surface of RuIr catalysts after CV cleaning. To verify this, firstly, we measured the XANES of RuIr catalysts at 1.20 V (after CV cleaning) (dotted red and yellow lines in Fig. 20e, see figure below). The white line (WL) positions of these spectra are located at lower energy than that of IrO_2 . This is because the inherent bulk average nature of XAFS experiments. The contribution of the metallic core dominated over the amorphous surface at 1.2 V. However, we struggled to analyze the XANES by using LCF method with three standards (Ir, K_3IrCl_6 and IrO_2). The spectra were not well-fitted, which suggested the existence of some other

species with oxidation states higher than 4+ (Figure A). Next, XANES spectra of the two catalysts at 1.65 V were measured. We found that the XANES at 1.65 V changes obviously compared to the ones at 1.2 V. To understand the change at the surface, we used a delta mu ($\Delta\mu$) method to isolate surface change by subtracting out the bulk information. The WL position of $\Delta\mu$ spectra of both RuIr catalysts is 1.3 V higher than the IrO₂. As for Ir L₃-edge, the increase of *d*-band hole is a function of the positive shift of WL position, that is 0.9 - 1.0 eV per *d*-band hole, which has been confirmed elsewhere (H. Nong et al., *Nat. Catal.* **1**, 841-851, 2018; C. Kim et al., *J. Am. Chem. Soc.*, **117**, 8557-8566, **1995**). With higher oxidation states (> 6+), this value might be slightly larger considering the covalence of bond. Therefore, the 1.3 eV shift could imply the existence of higher oxidation states (>4+) of Ir in the RuIr catalysts because Ir⁴⁺ species already exist in the surface of RuIr catalysts at 1.2 V. This result is consistent with that reported in Nature Catalysis 1, 841-851 (2018).

We added these data and analysis in the supplementary content and we also mentioned it in the main text accordingly.

Supplementary Fig. 20 e, Ir L₃-edge XANES spectra of the as-prepared RuIr catalysts at 1.20 V and 1.65 V along with Ir and IrO₂ as a comparison. f, Ir L₃ $\Delta\mu$ ($\mu_{1.65\text{ V}} - \mu_{1.20\text{ V}}$) spectra.

Figure A. LCF of the RuIr catalysts at 1.20 V just before OER. Here Ir³⁺ and Ir⁴⁺ are IrCl₃ and IrO₂, respectively.

Revision 1: Page 10, line 245 in the main text, we added “We also monitored the change of Ir L3-edge XANES during OER and found that the Ir in RuIr-NC is more resistant to oxidation compared that in RuIr-NS, which is similar to of the result of Ru K-edge (see detailed discussion in Supplementary Fig. 20e and f).”

Page 25-26, in the Supplementary, we added Fig. 20 e, f and the explanation as described above.

Reviewer #2:

The authors have addressed all my comments properly and the text has been modified accordingly.

The paper can be published as is.

Answer: Thank you for your recognition and many usefully suggestions in previous revision.

Reviewer #3:

The authors have made substantial effort to improving the manuscript and addressed the concerns raised by reviewers. However, there are still some improvements that should be made before the manuscript reaches the publication criteria of Nat. Commun.

Q1. There are still many grammatical errors to be corrected, even in the revisions made by the authors.

Answer: Thank you for pointing out this. To correct the grammatical errors, we had sent the draft to professional native grammar correction agency (<https://authorservices.springernature.com/>). We also rechecked our manuscript carefully to correct the grammatical errors. The change was list as follows.

Revision 2: Page 6, Line 153, “under potential” was change to “underpotential”.

Page 8, Line 209, “It has with similar crystal size” was change to “It has a similar crystal size”.

Q2. Response to the reviewer's second comment (Q2):

The revision made in Line 26, page 2 is not proper. 10 mA/cm²geo cannot be deemed as a high current density at all for an electrolyzer. The authors are recommended to revise the sentence as

“..., where current densities such as 10 mA/cm²geo can be readily achieved even at ...”

Note that “ready” is not the adjective form of “readily”. The authors should make clear the meanings of these words.

Answer: Thank you for your carefully checking. We changed the sentence as you suggested.

Revision 3: Page 2, Line 37, we revised the sentences accordingly.

Q3. Response to the reviewer's eighth comment (Q8):

The reviewer does not agree with the authors' logic. There are many improper practices and therefore unconvincing data in the literature. This should not become an excuse for all others to follow the incorrect practices in their research work. Two wrongs do not make a right. The authors should try their best to implement proper practices in their work, particularly for the work to be published in high-IF journals. Otherwise, more researchers will be misled in future. This misleading information will jeopardize the whole research community.

Revision: We appreciate your opinions and suggestion on this point. We totally agree with you and show the negatively scanned data in Fig. 2a and b in the main text, and positively scanned data in the Fig. 15d and Fig.23 in the Supplementary information as the references. Also, the value of the activities related to these figures were also corrected in the main text.

Revision 4: In the main text:

Page 4, Line 140: “**RuIr-NC** requires only 165 mV to achieve 10 mA cm⁻²_{geo} (Fig. 2a), showing much higher activity than Ir NPs (371 mV) and Ru NPs (550 mV), and the known optimal catalysts in acid (Supplementary Table 1). We must note that to avoid the overestimation of OER current density due to the metal oxidation current, we performed LSV from high potential to low potential (cathodic scan). Scan in a positive direction could exert a more serious oxidation/dissolution on the metals. However, the high OER activity/stability of **RuIr-NC** can be also reflected by the almost overlapped anodic and cathodic LSVs (Supplementary Fig. 13)”

Page 6, Line 150: “Specifically, at 1.45 V, **RuIr-NC** shows a j_m of 796 A g⁻¹_{metal} which is 2-4 times higher than the reported highly active catalysts”

Page 6, Line 154: “at 1.45 V, the **RuIr-NC** showed a j_s of 4.4 mA cm⁻²”

Page 7, Line 177: “At $10 \text{ mA cm}^{-2}_{\text{geo}}$, **RuIr-NS** requires an overpotential of **242 mV** (Fig. 2a), which is nearly **77 mV** higher than that of **RuIr-NC** (165 mV). However, when scan positively, **RuIr-NS** showed a quite similar or slightly higher current density than **RuIr-NC** before 1.4 V, and its current density did not increase as sharply as that of **RuIr-NC** thereafter (Fig. 13 a). This suggests the serious metal oxidation/dissolution in the **RuIr-NS**.”

Page 14: we changed Fig. 2a and b.

Fig. 2 | **OER performance.** **a**, Geometric activity (current density normalized by electrode surface area) of RuIr catalysts and pure Ru NPs and Ir NPs (LSV profiles are obtained by cathodic scan). **b**, Mass activity of **RuIr-NC**, **RuIr-NS** and the reported high-performance catalysts with considerable stability in acid including $\text{RuO}_2 \text{ NP}^{25}$, $\text{IrO}_2 \text{ NP}^{25}$, $\text{IrO}_x\text{-ATO}^{23}$, 3D Au-Ru^{28} , IrNiO_x^6 and $\text{Cr}_{0.6}\text{Ru}_{0.4}\text{O}_2^{27}$. The mass activities are normalized by the mass of noble metals. **c**, Chronopotentiometric curves under the OER current density of $1 \text{ mA cm}^{-2}_{\text{geo}}$. In the Supplementary information: We change Fig. 15 d and Fig.23 a and d by using the cathodic data.

Fig. 15d Specific activity (j_s) based on the EASA values together with the high-performance catalysts with considerable stability in acid.

Fig. 23 a, the geometric activity of RuIr-L. d, the specific activity of RuIr-L. RuIr-NC and RuIr-NS were used for comparison. Scan direction: cathodic.

Q4. Response to the reviewer's 12th & 13th comment (Q12 & Q13):

In Table S1, the authors only list the Ru or Ir based OER catalysts exhibiting performance poorer than that of their own RuIr-NC catalysts. This is not a proper practice. They should also make comparison with those having better performance than that of RuIr-NC, for example those reported in ACS Appl. Energy Mater. 2020, 3, 3746, where the nanoporous IrO₂ can sustain at 10 mA/cm² for 40 h as well and even at 100 mA/cm² for 30 h.

Revision: Thank you very much for your opinion. However, we have updated the results listed in both Table S1 and S4 with the paper you mentioned above and the other recently publications.

Revision 5: In the supporting information, the quite recent publications, Ref. 76-81 were added in Table S1 and Table S4.

Q5. Response to the reviewer's 16th comment (Q16):

The reviewer does not doubt that RuIr-L shows poorer performance compared to RuIr-NC. The reviewer's point is that the crystal size matters here and therefore the authors should revise their statement that "the crystal size is not a determined reason". The ECSA of RuIr-L is only 31.3 m²/g, nearly four times less than that of RuIr-NC (114 m²/g), how can the authors claim that they have "quite a similar ECSA"??

All statements must be made and supported by solid experimental data and theoretical analysis.

Answer: We are afraid that there might be a misunderstanding. **RuIr-L** is a large spherical particle which has similar crystal size, ECSA and atomic ratio to **RuIr-NC**. However, it shows lower activity and stability than **RuIr-NC**. As we shown in the supplementary Fig. 15c, the ECSA of **RuIr-NC** is 31 m²/g which is quite similar to that of **RuIr-L**. 114 m²/g is the ECSA of **RuIr-NS**. In order to avoid misunderstanding, we added the ECSA value of **RuIr-NC** in the supplementary.

Also, regarding to the statement “*the crystal size is not a determined reason*”, to make it clear, we changed the sentence as “**RuIr-NC** and **RuIr-NS** have different crystal size. However, the performance of RuIr-L could suggest that the crystal size might not be a determined reason within the study of interest.”

Revision 6: In the main text, Page 8, Line 209: “It has a similar crystal size, ECSA and atomic ratio to RuIr-NC (Supplementary Fig. 21, 22)”

Page 9, Line 212: “RuIr-NC and RuIr-NS have different crystal size. However, the performance of RuIr-L could suggest that the crystal size might not be a determined reason within the study of interest.”

In the Supplementary Fig. 15, we added “The ECSA of **RuIr-NC** is 31.0 m²/g”.

REVIEWERS' COMMENTS

Reviewer #1 (Remarks to the Author):

The authors have performed in situ results to support their findings. Therefore, I suggest this manuscript to be accepted in Nature Communications.

Reviewer #3 (Remarks to the Author):

The authors have now addressed all my concerns and I think the present manuscript can be accepted for publication as it is.